# Tissue-specific targeting of DNA nanodevices in a multicellular living organism

**Kasturi Chakraborty[1,2], Palapuravan Anees[1,2†], Sunaina Surana[1,2†], Simona Martin[1,2], Jihad Aburas[3], Sandrine Moutel[4,5], Franck Perez[5], Sandhya P Koushika[6], Paschalis Kratsios[2,3]\*, Yamuna Krishnan[1,2]\***

[1]Department of Chemistry, The University of Chicago, Chicago, United States; [2]Grossman Institute of Neuroscience, Quantitative Biology and Human Behavior, The University of Chicago, Chicago, United States; [3]Department of Neurobiology, The University of Chicago, Chicago, United States; [4]Recombinant Antibody Platform (TAb-IP), Institut Curie, PSL Research University, CNRS UMR144, Paris, France; [5]Cell Biology and Cancer Unit, Institut Curie, PSL Research University, CNRS UMR144, Paris, France; [6]Department of Biological Sciences, Tata Institute of Fundamental Research, Mumbai, India

**\*For correspondence:**
pkratsios@uchicago.edu (PK);
yamuna@uchicago.edu (YK)

†These authors contributed equally to this work

**Abstract** Nucleic acid nanodevices present great potential as agents for logic-based therapeutic intervention as well as in basic biology. Often, however, the disease targets that need corrective action are localized in specific organs, and thus realizing the full potential of DNA nanodevices also requires ways to target them to specific cell types in vivo. Here, we show that by exploiting either endogenous or synthetic receptor-ligand interactions and leveraging the biological barriers presented by the organism, we can target extraneously introduced DNA nanodevices to specific cell types in *Caenorhabditis elegans*, with subcellular precision. The amenability of DNA nanostructures to tissue-specific targeting in vivo significantly expands their utility in biomedical applications and discovery biology.

## Introduction

DNA has proven to be a versatile molecular scaffold to build an array of programmable synthetic nanoarchitectures due to its structural predictability (*Seeman and Sleiman, 2017*). The specificity of Watson–Crick–Franklin base pairing, its tunable affinity, the well-defined structural properties of the double helix, and its modular nature make the DNA scaffold highly engineerable. A wide array of functional DNA-based nanodevices have been deployed in vivo both for quantitative chemical imaging and also as programmable carriers that deliver encapsulated cargo upon receipt of a molecular cue (*Chakraborty et al., 2017*; *Chakraborty et al., 2016*; *Jani et al., 2020*; *Krishnan and Bathe, 2012*; *Lee et al., 2012*; *Li et al., 2011*; *Modi et al., 2009*; *Narayanaswamy et al., 2019*; *Saha et al., 2015*; *Sharma et al., 2014*; *Surana et al., 2011*; *Thekkan et al., 2019*; *Veetil et al., 2017*; *Zhao et al., 2012*). However, in most contexts, the site for payload action ideally needs to be confined to the desired organ or tissue. Thus, if DNA nanodevices can be targeted tissue-specifically in a live, multicellular organism, it would significantly expand their potential utility in biomedicine and fundamental biology.

Nature solves the problem of transporting poorly permeable molecules across membrane barriers, either releasing or enriching them tissue-specifically, by membrane trafficking. DNA nanostructures are particularly amenable to endosomal trafficking, which has in part led to an array of synthetic DNA

**Figure 1.** Schematic of strategies to target DNA devices to different cell types. (**a**) DNA nanodevices are intrinsically targeted to coelomocytes via the endogenously expressed scavenger receptors. (**b**) DNA nanodevices that display a dsRNA (green) domain are targeted to intestinal epithelial cells by engaging endogenously expressed SID-2 receptors. (**c**) DNA nanodevices are targeted selectively to neurons that express a DNA-binding protein (V$_H$H) fused to synaptobrevin-1 (*snb-1*). The DNA nanodevice has a sequence (blue) recognized specifically by V$_H$H.

nanostructures being presently deployed in living systems (*Bujold et al., 2018*; *Chakraborty et al., 2016*; *Krishnan et al., 2020*). Endosomal trafficking is ubiquitous: it regulates the internalization, sorting, and intracellular transport of diverse cargo, and is pivotal to development, signaling, and homeostasis. The significance and impact of endocytosis in health and disease is underscored by the observation that perturbations in endosomal trafficking are linked to multiple diseases, including neurodegeneration, cancer, and cardiovascular disease, and further, distinct cell types differ in their susceptibility to disease (*Maxfield, 2014*; *Mellman and Yarden, 2013*; *Mukherjee et al., 1997*). By leveraging membrane trafficking as well as taking advantage of biological barriers present in the whole organism, we show that DNA nanodevices can be targeted tissue-specifically and with organelle-level precision in the nematode *Caenorhabditis elegans*.

Most studies that use DNA nanostructures as reporters or payload carriers describe their functionality primarily in cultured cells that express scavenger receptors for which DNA is the natural ligand (*Li et al., 2011*; *Modi et al., 2009*). In vivo studies in multicellular organisms have also exploited scavenger receptor-mediated endocytosis to target DNA architectures to phagocytic cells such as coelomocytes in *C. elegans* (*Surana et al., 2011*) or microglia in *Danio rerio* (*Veetil et al., 2020*; *Figure 1a*). However, since most cells and tissues do not endogenously express scavenger receptors, targeting DNA nanostructures to such cell types in vivo remains challenging. In cultured cells, this problem is circumvented by exploiting naturally occurring receptor-ligand interactions. The natural ligand is chemically conjugated to a DNA nanodevice, which then binds its cognate receptor on the

cell surface and gets internalized and transported within the cell along the trafficking route adopted by the receptor (**Bhatia et al., 2016**; **Jani et al., 2020**; **Modi et al., 2013**). Alternately, a synthetic receptor-ligand interaction has been used where the receptor is a sequence-specific, DNA-binding protein based on a single-chain variable fragment (scFv) (**Modi et al., 2013**). When the scFv is fused to a trafficking protein such as furin, and the chimera is expressed in cells, furin displays the synthetic scFv receptor on the cell surface. Thus, DNA nanodevices with an scFv-binding sequence engage the scFv domain of the chimera and get trafficked to specific organelles within the cell (**Modi et al., 2013**; **Saminathan et al., 2021**). Despite these in cellulo demonstrations, there is still no evidence that DNA nanostructures can be targeted to tissues lacking scavenger receptors in multicellular organisms.

In this study, we demonstrate that DNA nanodevices can be targeted tissue-specifically in the nematode *C. elegans* by leveraging both natural and synthetic receptors on the surface of different cell types. We have focused on two tissues where endosomal trafficking has been linked to critical physiological functions, namely, the intestine and the nervous system. In the first instance, we exploit the presence of endogenous SID-2 receptors on the intestine to target a DNA nanostructure along the endo-lysosomal pathway in intestinal epithelial cells (IECs; **Figure 1b**). In the second, we present a generalizable route to target DNA nanostructures to cells lacking both scavenger receptors or SID-2 receptors. Here, we use a synthetic receptor, namely, a newly identified, recombinant, single-domain antibody (9E) that tightly binds a specific 4-nt sequence of dsDNA. When 9E is fused to the synaptic vesicle protein synaptobrevin-1 (SNB-1) and selectively expressed in neurons, the SNB-1::9E chimera binds DNA nanodevices having the cognate 4-nt domain and localizes them in trafficking endosomes in neurons (**Figure 1c**).

By leveraging the amenability of *C. elegans* to transgenesis, we demonstrate the molecular adaptability of this strategy. By expressing SNB-1::9E under promoters expressing in specific sets of neurons, we show that DNA nanodevices can be targeted to trafficking endosomes in specific neurons. We then demonstrate subcellular control over targeting afforded by this synthetic system. We further show that post-targeting, DNA nanodevices retain their functionality as reporters for chemical imaging. DNA nanodevices can be displayed on the neuronal surface without invoking their entry into organelles. When 9E is genetically fused to the transmembrane odorant receptor ODR-2 and expressed under a neuron-specific promoter, it binds and positions a DNA nanodevice on the neuronal surface. The small size, stability, and pH insensitivity of 9E make this two-component system immediately suitable for a multitude of DNA architectures bearing the cognate 4-nt sequence. Moreover, it offers a versatile route to target diverse proteins or track endosomal transport trajectories in different cell types in vivo, especially in model organisms amenable to transgenesis.

## Results and discussion
### Targeting DNA nanodevices to IECs

The intestine is one of the major organs of *C. elegans*, comprising a third of the total somatic mass with 20 large epithelial cells positioned with bilateral symmetry, forming a long tube around a lumen (**Altun and Hall, 2009**; **McGhee, 2007**). IECs contain prominent, birefringent gut granules that are known as lysosome-related organelles (LROs). Distinguished by lysosomal markers (**Coburn and Gems, 2013**; **Dell'Angelica et al., 2000**; **Hermann et al., 2005**), LROs play important roles in production and storage of melanin, immune defense, and neurological function (**Huizing et al., 2008**). We first sought to target a simple DNA nanodevice comprising a 38- bp double-stranded DNA (dsDNA) bearing a 5' Alexa-647 (A647), denoted $D^{38}$, to label LROs in the intestinal epithelia (**Supplementary file 1**). We reasoned that the biological barrier between the intestinal lumen and the pseudocoelom would effectively preclude $D^{38}$ from access to coelomocytes, as seen whenever DNA nanodevices are microinjected in the pseudocoelom (**Bhatia et al., 2011**; **Surana et al., 2011**).

We therefore utilized a liquid-feeding method to introduce 1 μM of $D^{38}$ into the intestinal lumen of 1-day adult worms (M&M). We leveraged the high autofluorescence of LROs at shorter wavelengths to regions of interest (ROIs) to evaluate the efficacy of $D^{38}$ uptake. The mean intensity corresponding to $D^{38}$ uptake, compared to control mock-fed worms, was very low, indicating low or no intrinsic targetability of DNA nanodevices to IECs (**Figure 2a,b**). We therefore varied the size of the DNA nanodevice to test whether increasing the number of base pairs (bp) and/or negative charges improved targeting, the sequences of which are shown in **Supplementary file 1**. Using the same liquid-feeding method,

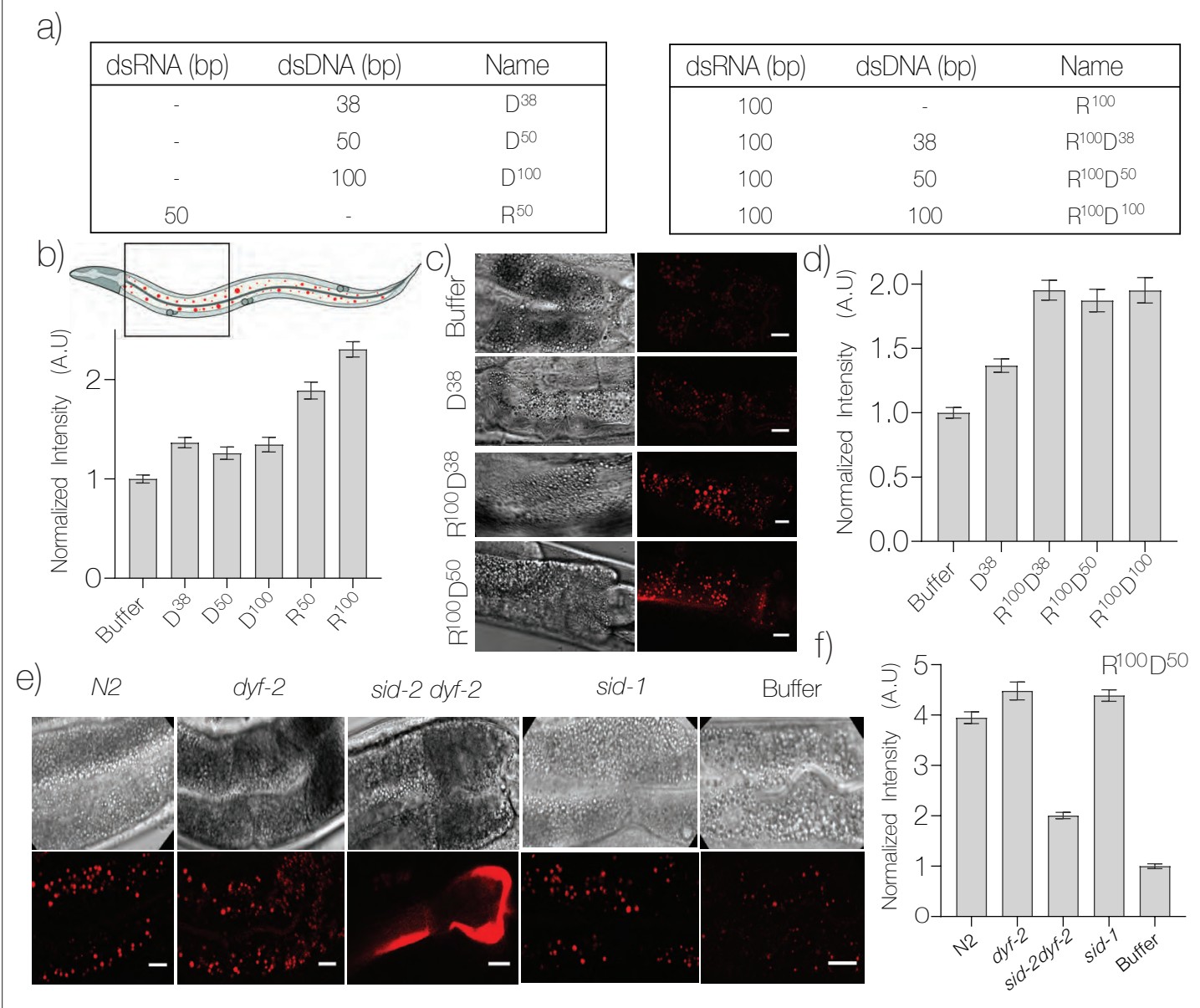

**Figure 2.** DNA nanodevices are targeted to intestinal epithelial cells (IECs). (**a**) Table of the composition and length of the various Alexa647 labeled DNA nanodevices used. (**b**) Mean Alexa647 fluorescence intensity corresponding to the uptake of the indicated nanodevice in *C. elegans* IECs. (**c**) Representative fluorescence and DIC images of IECs labeled with the indicated R$^{100}$-conjugated nanodevice quantified in (**d**). (**e**) Representative fluorescence images of IECs labeled with nanodevice R$^{100}$D$^{50}$ in the indicated genetic background quantified in (**f**). Data represented as mean ± standard error of the mean of 10 worms, ~500 endosomes. The four IECs closest to the pharynx were considered for quantification. Scale bar = 10 μm.

The online version of this article includes the following source data and figure supplement(s) for figure 2:

**Source data 1.** Quantitation of uptake of nanodevices by IECs.

**Source data 2.** Quantitation of uptake of R100 conjugated nanodevices by IECs.

**Source data 3.** Quantitation of uptake of R100D50 nanodevices by IECs in various genetic backgrounds.

**Figure supplement 1.** Uptake of nucleic acid probes in the intestinal epithelial cells.

**Figure supplement 1—source data 1.** Pearson's correlation coefficient (PCC) for colocalization between Rn Dn and LRO markers.

we repeated uptake experiments with 50- or 100- bp-long dsDNA ($D^{50}$ or $D^{100}$). Again, there was no enhancement in nanodevice targeting to IECs (*Figure 2a*). This revealed that DNA nanostructures are not inherently targeted to cells in this tissue.

We then tested whether we could co-opt the RNA interference (RNAi) pathway in order to target DNA-based probes to intestinal epithelia. Previous studies have shown that worms are capable of endocytosing dsRNA localized in the acidic intestinal lumen, and that this is effected by the membrane protein SID-2 (*Hunter et al., 2006*; *McEwan et al., 2012*; *Winston et al., 2007*). We first tested the uptake of dsRNAs either 50- bp or 100- bp long, denoted $R^{50}$ or $R^{100}$, respectively, based on previous work suggesting this as the minimum length needed for effective internalization by SID-2 (*Figure 2a*; *McEwan et al., 2012*). All dsRNAs carried a fluorescent Alexa647 label at their 5′ termini. Interestingly, when worms were liquid-fed either $R^{100}$ or $R^{50}$ we observed that uptake efficiency by intestinal epithelia increased approximately two fold compared to their DNA analogues (*Figure 2b*).

We therefore sought to test whether covalently attaching $R^{100}$ or $R^{50}$ to a DNA nanodevice would promote nanodevice entry into the intestinal epithelia via the SID-2 receptor, and potentially label LROs. Using click chemistry, we conjugated $R^{100}$ to DNA nanodevices $D^{38}$, $D^{50}$, or $D^{100}$ bearing Alexa647 labels to give chimeric RNA-DNA nanodevices denoted $R^{100}D^{38}$, $R^{100}D^{50}$, and $R^{100}D^{100}$, respectively (*Figure 2c,d*, *Figure 2—figure supplement 1*; *Jewett et al., 2010*). When nematodes were liquid-fed the above chimeras and then imaged, we found that overall conjugating $R^{100}$ to DNA nanodevices improved uptake of the latter by IECs. However, interestingly, we observed that the length of the dsDNA scaffold had no significant effect on uptake efficiency (*Figure 2d,e*). Thus, it is possible to target DNA nanodevices to IECs using a DNA-RNA chimera.

Closer scrutiny revealed that $R^{50}D^{n}$ nanodevices internalized by intestinal epithelia localized to LROs, readily identified by their high autofluorescence at low wavelengths (*Soukas et al., 2013*), as well as colocalization with LAMP-1::GFP and GLO::1-GFP (*Figure 2—figure supplement 1*; *Schroeder et al., 2007*; *Soukas et al., 2013*). In order to confirm the identity of the receptor responsible for the uptake of the chimeric nanodevices into LROs, we repeated uptake assays in worms carrying mutant alleles for *sid-1* and *sid-2*, two critical players of the RNAi pathway. Homozygous animals for the *sid-1(qt9)* null allele are systemically resistant to RNAi (*Winston et al., 2002*). The *sid-2* locus is embedded within an intron of another gene (*dyf-2*), and the only available strong loss-of-function allele (*gk505*) for *sid-2* contains a 403- bp deletion, which removes the first *sid-2* exon and the 12th *dyf-2* exon. As a control for the *dyf-2&ZK520.2(gk505)* strain (referred to here as, *sid-2&dyf-2*), we used animals carrying the *dyf-2(gk678)* mutant allele that selectively disrupts the *dyf-2* gene . Wild-type (N2 strain), *dyf-2*, *sid-1*, and *sid-2&dyf-2* worms were tested for their ability to uptake $R^{100}D^{50}$ as previously described (*Figure 2d,e*). We found that uptake was reduced in *sid-2&dyf-2* mutants, but not in *sid-1(qt9)*, *dyf-2(gk678)*, or wild-type worms. Our results are consistent with SID-2 being localized on the extracellular surface, acting as a receptor that binds and endocytoses dsRNA, while SID-1 is resident in endosomes, where it binds dsRNA cargo and transports it into the cytosol (*Jose and Hunter, 2007*; *Li et al., 2015*; *McEwan et al., 2012*). This implicates SID-2 as the receptor for nanodevice internalization and its subsequent trafficking to LROs in IECs.

## Development of a DNA-binding recombinant humanized V$_H$H

*C. elegans* has proven to be a powerful model organism to study vesicular trafficking in neurons, given its transparent body and simple nervous system (*Hulme and Whitesides, 2011*; *Teschendorf and Link, 2009*). However, neurons do not express either scavenger receptors or SID proteins, complicating the targeting of DNA nanodevices to these cell types. We therefore developed a DNA-binding, recombinant, humanized V$_H$H as a synthetic receptor for DNA nanodevices. This offers the advantage of tagging endogenously expressed proteins in any cell of choice rather than ectopically expressing canonical DNA-binding proteins that could have unknown biological effects. The small size of these antibodies, the ability to select them by display technologies, and tune their affinity, stability, and expression by molecular evolution have led to a plethora of in vivo applications (*Winter et al., 1994*). Humanized V$_H$H antibodies lack a light chain altogether, and the heavy chain itself is sufficient to form the antigen-binding pocket (*Hamers et al., 1993*), leading to their utility in therapeutics as well as cell biology (*Harmsen and De Haard, 2007*). They robustly maintain their native conformations due to increased hydrophilicity and single-domain nature and are much more resistant to thermal and

chemical denaturation (*Dumoulin et al., 2002*; *Ewert et al., 2002*; *Pérez et al., 2001*; *van der Linden et al., 1999*).

We screened a recombinant humanized $V_H$H antibody library to isolate high-affinity candidates that bind a specific DNA duplex and characterized the binding in vitro (*Moutel et al., 2016*). After a phage display screen, we studied 160 potential binders and selected one of the highest affinity binders for further evaluation for its affinity and specificity of binding. This binder, 9E, was further tested for its expression, pH dependence, and binding affinity. To date, $V_H$H recombinant antibodies have been obtained using phage display against many classes of molecules, but not against nucleic acids.

We developed an assay to screen recombinant humanized $V_H$H antibodies by phage display against a 41- bp dsDNA to obtain high-affinity, sequence-specific DNA-binding antibodies (Materials and methods). The 41 -bp dsDNA target was immobilized on streptavidin-coated magnetic beads via a 5′ biotinylated terminus and presented as the epitope to the $V_H$H library (*Figure 3a*). Following standard procedure , three rounds of progressive selection and amplification enriched for putative dsDNA binders. Next, 160 clones were randomly selected, grown in 96-well plates, and the corresponding phage bound $V_H$H antibodies were expressed and screened for dsDNA binders. Nearly 70 % displayed DNA-binding properties, of which 40 bound DNA regardless of whether it was double- or single-stranded. Interestingly, 42 were found to show sequence specificity for the dsDNA target and showed minimal binding to ssDNA of the same sequence or dsDNA with a different sequence. These 42 clones were taken forward for further analysis (*Figure 3—figure supplement 1*).

To screen for the minimal dsDNA motif recognized by each of the selected $V_H$H antibody clones, ELISA was performed against a set of immobilized dsDNAs. These corresponded to the full 41-mer dsDNA target and three shorter dsDNA regions $R_1$, $R_2$, and $R_M$ on the target (*Figure 3b*). Region $R_1$ corresponded to a 17 -bp region at the 3′ terminus of the biotinylated ssDNA oligonucleotide (nucleotides in red and blue font), $R_2$ corresponded to a 17 -bp region on the 5′ end of the oligonucleotide (nucleotides in green font), while $R_M$ corresponded to a 17 -bp overlapping the 5′ end of $R_1$ and the 3′ of $R_2$ shown in italics. Of the 42-dsDNA-binding $V_H$H antibodies tested, nearly 80 % specifically bound region $R_1$, albeit with varying affinities (*Figure 3—figure supplement 1*).

To pinpoint the epitope on the 17 -bp region $R_1$, partial duplexes were tested covering this region with a sliding window of 4- bp from the 3′ end of the biotinylated oligonucleotide to form dsDNAs $R_3$ and $R_6$ (*Figure 3b*). This strategy ensured that the epitope could be narrowed down to at least 4- bp. ELISAs revealed that each of the 42 $V_H$H antibodies required only the first 4- bp at the 3′ end of region $R_1$ (ATAA, nucleotides in red) for binding (*Figure 3c*). When this 4 -bp sequence is present not at the terminus, but in the middle of the dsDNA duplex, as in $R_6$, it was no longer a ligand for any of the 42 $V_H$H antibodies. This clearly demonstrated that the minimal binding epitope for all 42 $V_H$H antibodies isolated was a terminal d(ATAA) motif (*Figure 3—figure supplement 2*). Furthermore, since endocytic events are accompanied by lumenal pH changes, it is crucial to test whether binding between the $V_H$H antibody to its cognate DNA epitope is pH-independent. From a pH-dependent ELISA screen, we identified that the $V_H$H 9E was disrupted the least by acidification, demonstrating ~82 % binding even at pH 5 (*Figure 3d*, *Figure 3—figure supplement 3*). Finally, based on ELISA, we estimated that the relative binding affinity of $V_H$H 9E to its target epitope was ~200 nM (*Figure 3e*, Materials and methods). Cumulatively, our findings show that the recombinant, humanized $V_H$H antibody 9E binds with high specificity to a 3′ terminal, d(ATAA) region (*Figure 3—figure supplement 4*).

## Pan-neuronal expression of the SNB-1::9E chimera in *C. elegans*

Next, we tested whether DNA nanodevices could be targeted to neuronal endosomes by genetically fusing 9E, the DNA-binding $V_H$H antibody identified above, with a protein such as synaptobrevin-1 (SNB-1), that is expressed in neurons and undergoes endosomal trafficking. Our choice of synaptobrevin-1 as a carrier protein was guided by it being an integral membrane protein present in multiple copies in synaptic vesicles (*Dittman and Ryan, 2009*). It has a low molecular weight (~16 kDa) and its N-terminal cytoplasmic domain interacts with target membrane SNARE (t-SNARE) proteins to facilitate neurotransmitter exocytosis, while its C-terminus has a hydrophobic patch that anchors it to vesicular membranes. The C-terminus ends in a short tail that extends into the vesicular lumen and is exposed to the synaptic cleft during exocytosis (*Hanson et al., 1997*; *Südhof, 1995*). Upon their fusion with the plasma membrane at the synaptic cleft, synaptic vesicles undergo recycling via the endosomal pathway (*Rizzoli, 2014*). Furthermore, fusing GFP (~27 kDa) to the C-terminus of SNB-1

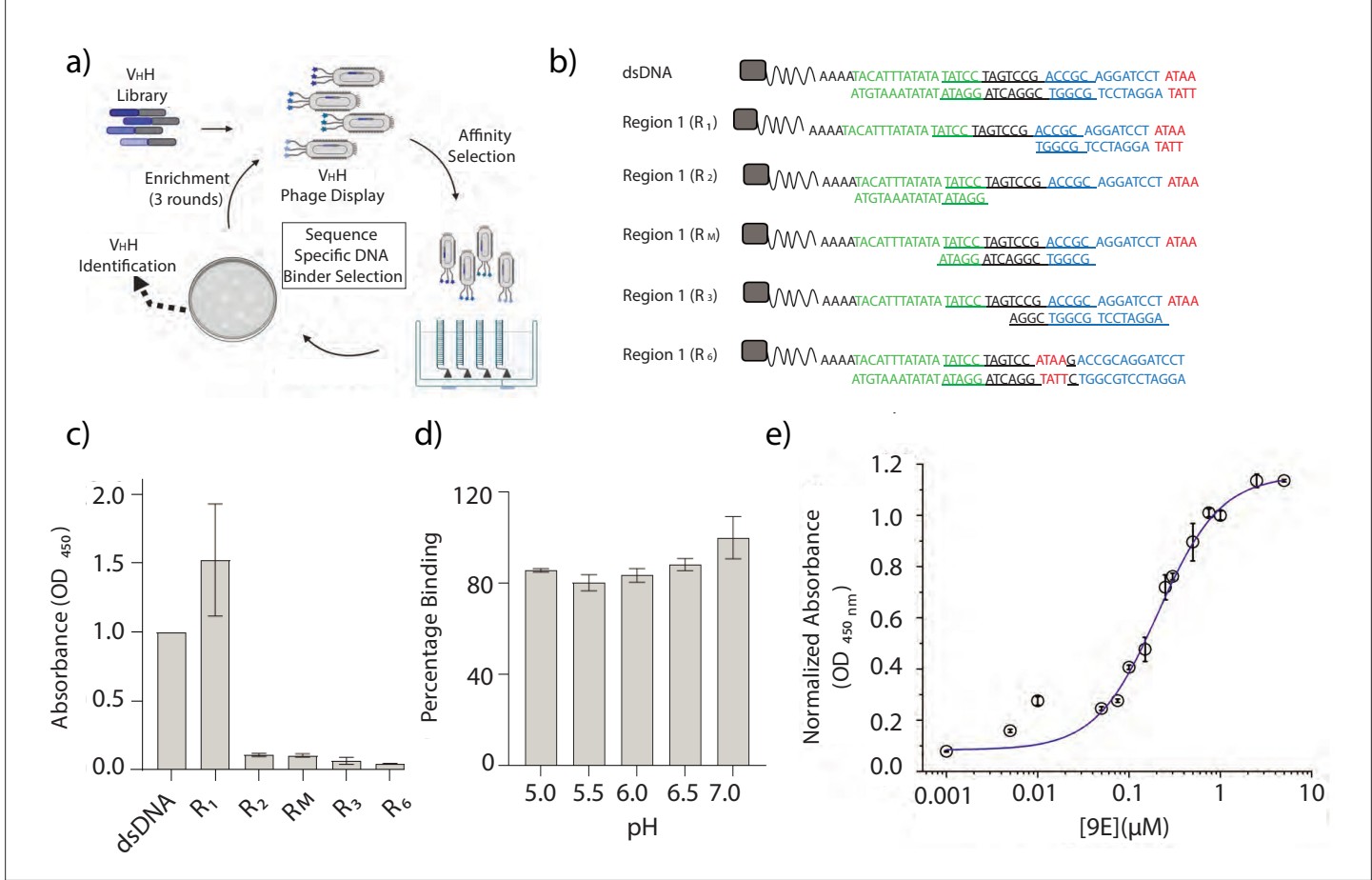

**Figure 3.** Identification and characterization of a sequence-specific DNA-binding recombinant antibody 9E. (**a**) Schematic of phage display screen to identify DNA binders using a humanized V$_H$H antibody library. (**b**) Sequence of the various dsDNA epitopes used to pinpoint the dsDNA sequence bound by the recombinant antibody, 9E. The tested regions are R$_1$ (blue and red), R$_2$ (green), an overlapping region R$_M$ (italicized and underlined), R$_3$ (region R1 frame-shifted by a window of 4 nt), and R$_6$ (4-nt motif in red, present in the middle of the DNA duplex). Biotin (gray square) was incorporated to immobilize the DNA epitopes on streptavidin-coated magnetic beads. (**c**) Binding efficiencies of the recombinant antibody 9E with the indicated dsDNA constructs as determined by ELISA. (**d**) Effect of pH on binding efficiency of 9E with dsDNA as determined by ELISA. (**e**) The relative binding constant of 9E and dsDNA epitope in solution. Serial dilutions of the purified protein were added to a fixed amount of immobilized dsDNA (25 pmoles). All experiments were performed in triplicate, and the data is represented as mean ± s.e.m.

The online version of this article includes the following source data and figure supplement(s) for figure 3:

**Source data 1.** Binding efficiencies of the recombinant antibody 9E with dsDNA.

**Source data 2.** Effect of pH on binding efficiencies.

**Source data 3.** Relative binding constant of 9E and dsDNA.

**Figure supplement 1.** Analysis of recombinant antibody binders of adsDNA epitope.

**Figure supplement 1—source data 1.** Yield of the phages after each round of selection.

**Figure supplement 2.** Characterization of dsDNA-binding V$_H$H antibodies.

**Figure supplement 2—source data 1.** Relative binding of V$_H$H antibodies to duplexes R$_1$, R$_2$, and R$_M$.

**Figure supplement 2—source data 2.** Relative binding of V$_H$H antibodies to duplexes R$_3$,R$_4$, and R$_5$.

**Figure supplement 2—source data 3.** Relative binding of V$_H$H antibodies to duplex R$_6$.

**Figure supplement 3.** Characterization of 9E.

**Figure supplement 3—source data 1.** Determination of affinity of 9E for dsDNA_1.

**Figure supplement 3—source data 2.** Determination of affinity of 9E for dsDNA_2.

**Figure supplement 4.** Electrophoretic mobility shift assay (EMSA) to demonstrate binding of 9E to the 4-nt minimal binding motif.

(*Murthy et al., 2011*; *Nonet, 1999*) does not perturb SNB-1 localization or function (*Nonet, 1999*; *Figure 1c*).

Therefore, we reasoned that neuronal expression of an SNB-1::9E fusion protein would lead to the 15 kDa $V_H$H domain being displayed in the synaptic cleft upon neurotransmitter release. Thus, if DNA nanodevices bearing a terminal d(ATAA) motif were present in the pseudocoelom, they could bind the $V_H$H domain (blue) of the chimera at the synaptic cleft and get trafficked along the endosomal pathway (Scheme 1). We therefore generated transgenic *C. elegans* animals, denoted p*snb-1::snb-1::9E* worms, that express the SNB-1::9E chimera in all neurons under the control of the *snb-1* promoter (p*snb-1*) (*Stefanakis et al., 2015*), including an *unc-54* 3' UTR for efficient translation (*Figure 4a,b*, *Figure 4—figure supplement 1*). Into these worms we injected the DNA nanodevice $D^{38}$ (1 µM) modified to display a terminal d(ATAA) motif and an ATTO 647 N fluorophore, now denoted $nD^{A647N}$, into the pseudocoelom and imaged the worms after 30 min. We observed several punctate structures, linearly arranged along the worm's body strongly resembling neuronal synapses along the *C. elegans* dorsal and ventral nerve cords (*Figure 4b*).

To test whether the nanodevices were in fact labeling neurons and synapses, we performed a series of colocalization experiments using transgenic *C. elegans* animals carrying different neuronal and synaptic markers. The strain *otIs355 [rab-3p(prom1)::2xNLS::TagRFP]* expresses nuclear RFP pan-neuronally, whereas *otIs45 [unc-119::GFP]* expresses cytosolic GFP pan-neuronally (*Figure 4—figure supplement 2*; *Altun-Gultekin et al., 2001*; *Nguyen et al., 2016*). The *jsIs682 [rab-3p::GFP::rab-3 + lin-15(+)]* strain expresses GFP::RAB-3 in most neurons, which is localized primarily to synaptic regions (*Figure 4—figure supplement 2*; *Mahoney et al., 2006*). Hermaphrodites carrying the p*snb-1::snb-1::9E* transgene were crossed with each of the aforementioned reporter strains (*Figure 4—figure supplement 1*). When 500 nM $nD^{A647N}$ was injected into *jsIs682 [rab-3p::GFP::rab-3 + lin-15(+)]*; p*snb-1::snb-1::9E* worms, we found that $nD^{A647N}$ labeling coincides with GFP::RAB-3, confirming synaptic labeling. Interestingly, $nD^{A647N}$ containing puncta do not exclusively colocalize with GFP::RAB-3-positive puncta within the same axon. To evaluate colocalization, we plotted line profiles for ROIs drawn perpendicular to long axis of different neurons, which revealed ~80 % colocalization (*Figure 4—figure supplement 2*). This suggests that in addition to the synaptic regions of neurons, $nD^{A647N}$ might be labeling trafficking endosomal compartments or synaptic vesicles that are being recycled. Similar colocalization experiments of $nD^{A647N}$ with other neuronal markers such as *otIs355 [rab-3p(prom1)]::2xNLS::TagRFP* and *otIs45 [unc-119::GFP]* further confirmed that $nD^{A647N}$ indeed labeled neurons (*Figure 4—figure supplement 2*).

Normally when DNA nanodevices are introduced into the pseudocoelom, they are taken up by scavenger receptors present on coelomocytes (*Chakraborty et al., 2017*; *Dan et al., 2019*; *Narayanaswamy et al., 2019*; *Surana et al., 2011*; *Veetil et al., 2017*). A key characteristic of nanodevice uptake via this pathway is that uptake is easily abolished in the presence of 10 equivalent molar excess of maleylated BSA as the latter competes for scavenger receptors due to its anionic nature (*Haberland and Fogelman, 1985*). Importantly, even in the presence of 10 equivalents excess of maleylated BSA, neuronal labeling of p*snb-1::snb-1::9E* worms with $nD^{A647N}$ was not affected (*Figure 4c–i*). This indicates that $nD^{A647N}$ uptake into neurons does not occur via scavenger receptors. When the terminal d(ATAA) motif in $nD^{A647N}$ was removed to give a DNA nanodevice denoted $nD2^{A647}$, we observed no such neuronal labeling (*Figure 4cii*). These results show that DNA nanodevices can be targeted to neurons in worms only if the former incorporate a d(ATAA) motif and if the latter express the SNB-1::9E chimera in neurons.

## Targeting DNA nanodevices with cellular and subcellular precision

To assess the precision over targeting using our method, we sought to localize DNA nanodevices in endosomes of specific neuron types. We therefore expressed the SNB-1::9E chimera under two different neuronal promoters. We focused on the cholinergic subset of motor neurons since acetylcholine is the most broadly used neurotransmitter in the nematode nervous system (*Pereira et al., 2015*). Cholinergic motor neurons in the ventral nerve cord are divided into six classes: DA, DB, AS, VA, VB, and VC (*Kerk et al., 2017*). Since the acetylcholine receptor subunit ACR-2 is expressed in four of the six classes of cholinergic motor neurons (DA, DB, VA, VB), we used the *acr-2* promoter region to target these subsets of neurons (*Kratsios et al., 2011*). Transgenic p*acr-2::snb-1::9E* worms were generated using the previously described *pha-1* strategy (*Figure 4a*; *Granato et al., 1994*). When $nD^{A647N}$ (1 µM)

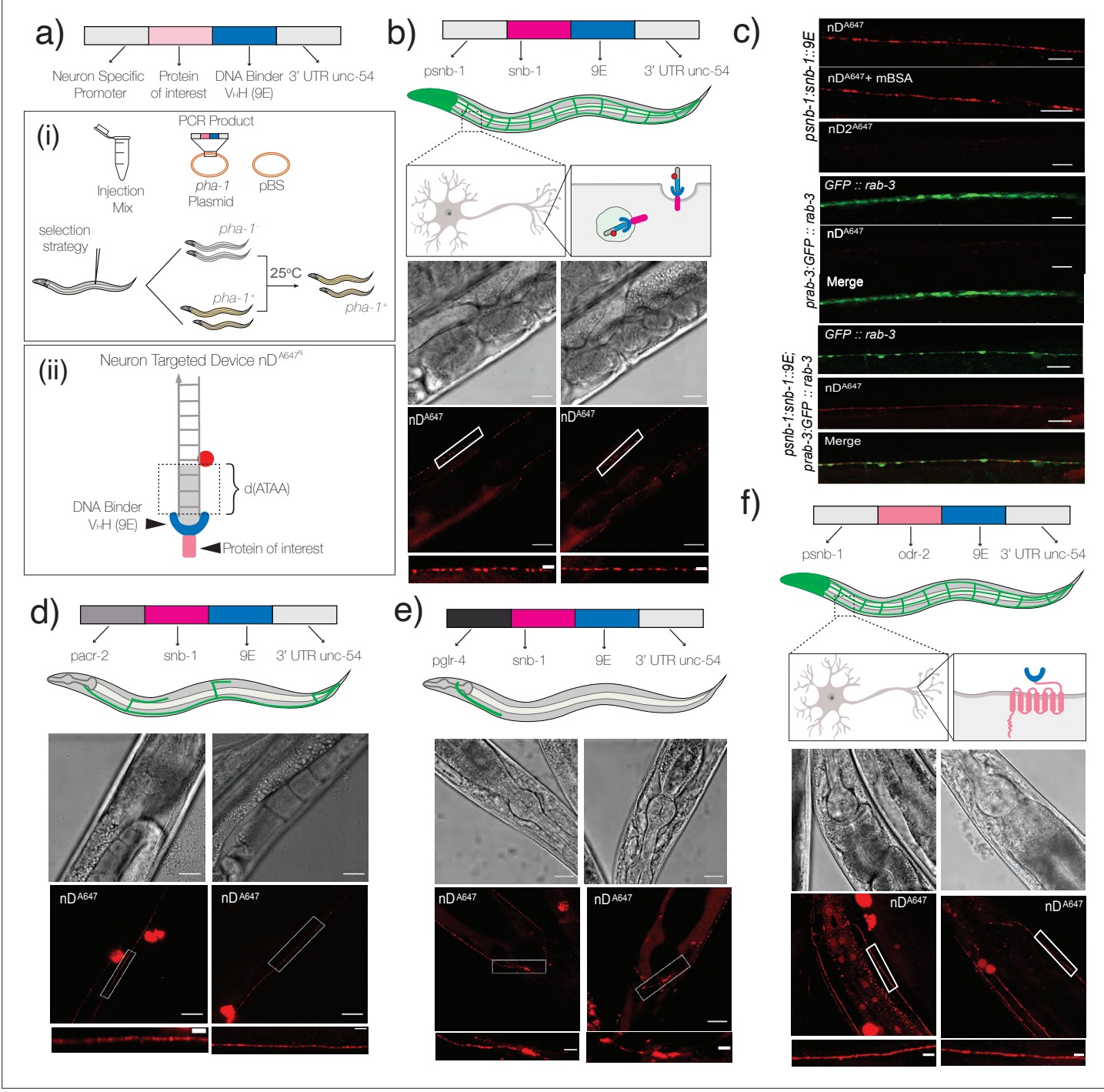

**Figure 4.** Targeting DNA nanodevices to neurons. (**a**) Schematic of the constructs used to make transgenics: (**i**) strategy to select transgenics based on *pha-1+* worms; (**ii**) schematic of neuron targetable DNA nanodevice nD^A647N bound to its synthetic receptor, 9E, fused to the protein of interest. (**b**) Schematic of nanodevice uptake into neurons of p*snb-1::snb-1::9E* worms. Brightfield and fluorescence images of *C. elegans* neurons labeled with nD^A647N. (**c**) Representative fluorescence images of (i) neurons in p*snb-3::snb-1::9E* worms injected with nD^A647N in the presence or absence of mBSA and nD2^A647 lacking the 3′ terminal d(ATAA); (ii) neurons in p*rab-3::gfp::rab-3;* and (iii) p*rab-3::gfp::rab-3;* p*snb-1::snb-1::9E* expressing *C. elegans* injected with nD^A647N. nD^A647N can be targeted to neuronal subsets. (**d**) nD^A647N labels cholinergic motor neurons in p*acr-2::snb-1::9E* and head neurons in (**e**) p*glr-4::snb-1::9E* expressing *C. elegans*. nD^A647N can be targeted via the labeling cassette to other proteins cell-specifically. (**f**) Images of neurons in p*snb1::odr-2::9E* expressing animals labeled with nD^A647N. In all images, the white boxed region is shown in the zoomed image below. Scale: 20 µm; Inset: 5 µm.

The online version of this article includes the following source data and figure supplement(s) for figure 4:

**Figure supplement 1.** Gel electrophoresis characterization of PCR fragments and transgenic worms.

*Figure 4 continued on next page*

Figure 4 continued

**Figure supplement 2.** Representative images of transgenic worms containing both (**a**) *rab-3p(prom1)::2xNLS::TagRFP* or (**b**, **c**) *unc-119::GFP* or (**d**) p*rab-3::gfp::rab-3* and p*snb-1::snb-1::9E* injected with nDNA.

**Figure supplement 2—source data 1.** Line intensity profiles for neuron 1.

**Figure supplement 2—source data 2.** Line intensity profiles for neuron 2.

was introduced into the pseudocoelom of these nematodes, we observed that it localized in several, linearly arranged punctate structures resembling synapses and/or neuronal endosomes, around the mid-section of the worm body corresponding to the axons of the VA and VB neurons of the ventral nerve cord (*Figure 4d*). The DA and DB neurons extend their axons to the dorsal nerve cord and form synapse with dorsal muscle. However, we did not observe any labeling with nD$^{A647N}$ in the dorsal nerve cord of p*acr-2::SNB-1::9E* worms. Thus, our strategy selectively labels VA and VB neurons over DA and DB classes of neurons. Such selective labeling could arise from the differential cell-surface levels of SNB-1::9E in VA and VB neurons, or greater accessibility of their synapses to cargo in the pseudo-coelom, or both.

Next, we targeted head neurons and a distinct set of cholinergic motor neurons, namely the SAB class, located proximal to the pharynx that innervates the head muscles. In order to target nD$^{A647N}$ to these neurons, we chose the promoter of the glutamate receptor family protein GLR-4 (*Feng et al., 2006*) since this protein is known to be expressed in the head neurons and the SAB, and DA9 neurons (*Brockie et al., 2001*; "glr-4 (gene) - WormBase: *Nematode Information Resource, 2021*; *Hills et al., 2004*; *Kratsios et al., 2015*; *Rothaug et al., 2014*). When nD$^{A647N}$ (1 μM) was introduced into the pseudocoelom of these nematodes, we observed that it localized in punctate structures arranged around the pharynx, corresponding to the putative synapses of head neurons expressing GLR-4, and ventral neuromuscular synapses belonging to the SAB neurons (*Figure 4e*). Again, just as in the previous case of the DA and DB neurons, the DA9 neurons at the tail of the worm did not show labeling, revealing that this method led to the preferential labeling of SAB neurons. Taken together, our results show that DNA nanodevices can be targeted to synapses and/or trafficking endosomes of specific neuron types expressing SNB-1::9E.

We then tested whether our method provides subcellular-level precision in terms of targeting individual neurons. Therefore, we tested whether we could target a DNA nanodevice to neurons, and yet not allow neuronal entry, thereby keeping the DNA nanodevice immobilized on the neuronal surface. We therefore genetically fused 9E to the ODR-2 protein and expressed it under the pan-neuronal p*snb-1* promoter (*Figure 4f*). ODR-2 is a GPI-anchored, membrane-associated signaling protein that plays important roles in nematode chemotaxis (*Chou et al., 2001*). ODR-2 is highly expressed in sensory neurons, motor neurons, and interneurons, and is particularly enriched in axons (*Gottschalk and Schafer, 2006*). Using the *pha-1* selection strategy described earlier, we obtained transgenics stably expressing the ODR-2::9E chimera (*Figure 4a*). When these transgenics were injected with nD$^{A647N}$ (1 μM), we observed that in all worms the DNA nanodevice labels neuronal axons corresponding to the ventral and the dorsal nerve cords (*Figure 4f*). In these transgenics, a near-uniform pattern of plasma membrane labeling was observed that is reminiscent of ODR-2 staining (*Chou et al., 2001*), and is in stark contrast to the punctate labeling that was observed in p*snb-1::snb-1::9E* worms (*Figure 4b*). Taken together, our experiments show that the synthetic receptor-ligand strategy shown here is generalizable to cell-surface membrane proteins and can be used to label specific membrane domains and/or endosomal compartments across neurons in vivo. Additionally, this design is applicable to a range of functional DNA nanostructures provided they harbor a d(ATAA) tag, opening up new avenues for the cell-specific application of DNA nanodevices in biological systems.

To demonstrate application of a functional nanodevice to sensing local environments in neurons, we used *pHlava-9E,* a modified version of a DNA-based pH reporter developed recently that uses pHrodo(O) (*Veetil et al., 2020*). To enable ratiometric quantification, *pHlava-9E* contains pH-insensitive dye, ATTO 647 N (R), on one of its strands. To enable internalization by 9E, *pHlava-9E* includes a terminal d(ATAA) sequence. Device formation was characterized by gel electrophoresis (*Figure 5—figure supplement 1*). *pHlava-9E* is designed to provide a digital output of compartmentalization, that is, its pH profile is such that even if it is internalized into a mildly acidic vesicle, the pH readout is as high as one would observe with a lysosome. The in vitro pH performance curve of *pHlava-9E* shows a maximum fold change in O/R of 2.5 from pH 8 to pH 6, indicating that *pHlava-9E* is particularly

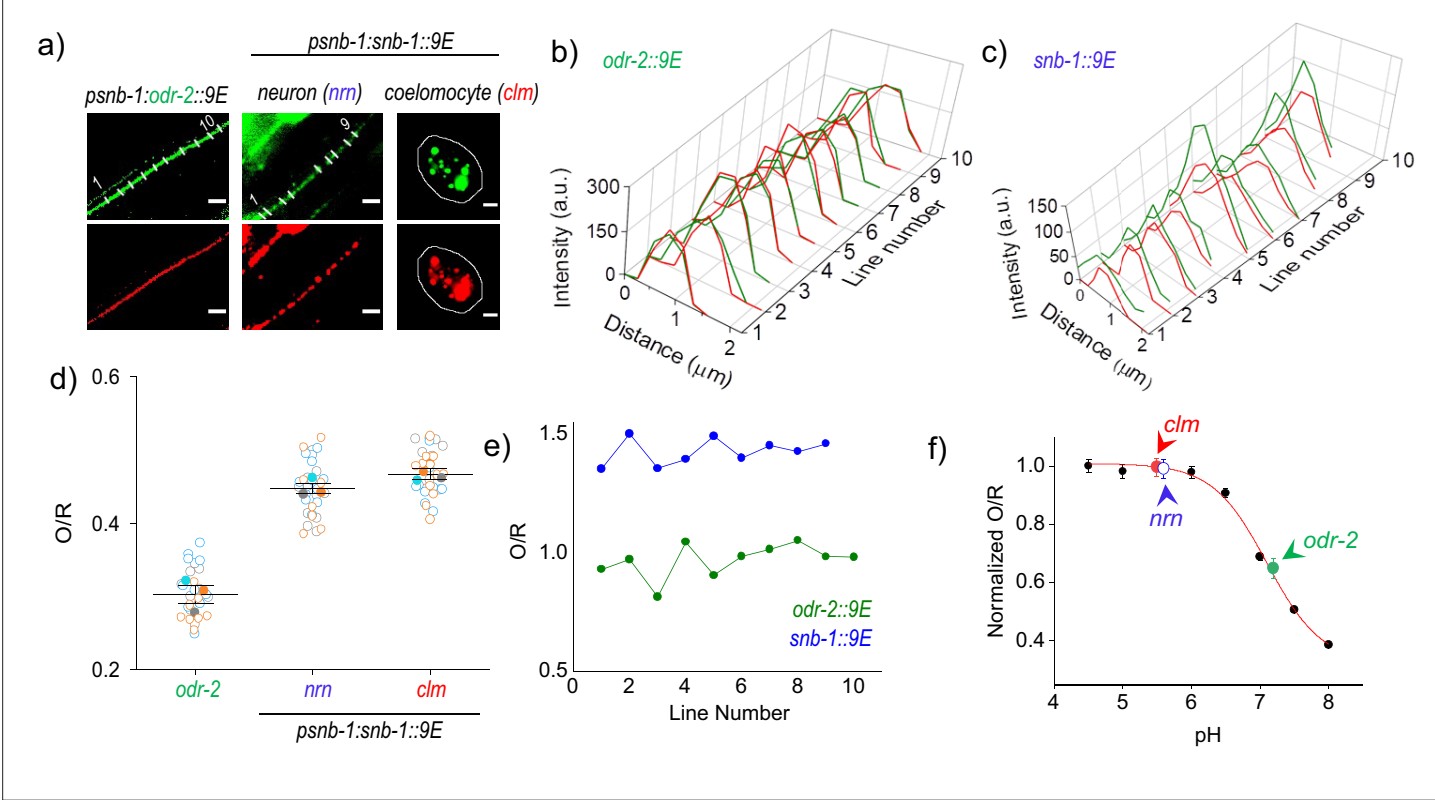

**Figure 5.** Mapping the pH of cell-surface versus internalized nanodevices using *pHlava-9E*. (**a**) Representative images of *pHlava-9E* labeled compartments in neurons (nrn) and coelomocytes (clm) of p*snb-1::snb-1::9E* worms and neurons in p*snb-1::odr-2::9E*, acquired in the TMR (O) and ATTO 647 N (R) channels. Scale bar, 5 µm. (**b, c**) Pixel-wise line intensity profiles in the O (green lines) and R (red lines) channels of *pHlava-9E* labeled neurons in (**b**) p*snb-1::snb-1::9E* worms and (**c**) p*snb-1::odr-2::9E* along each indicated white line in (**a**). (**d**) O/R values reported by *pHlava-9E* along each indicated line in (**a**). (**e**) Distributions of O/R values from three representative datasets for each of the three regions, one of which is shown in (**a**). Data points from each dataset are shown in a unfilled circles of different colors, and the color-filled circles correspond to the average of that dataset. Lines depict average and standard error of the mean. (**f**) pH calibration profile of *pHlava-9E* showing normalized TMR and Alexa 647 fluorescence intensity ratio (O/R) versus pH, on which is overlaid the O/R values of *pHlava-9E* labeled neurons and coelomocytes in p*snb-1:snb-1::9E* and neurons in p*snb-1:odr-2::9E* worms. Error bars indicate the mean of three independent experiments ± s.e.m.

The online version of this article includes the following source data and figure supplement(s) for figure 5:

**Source data 1.** Pixel-wise line intensity profiles of *pHlava-9E* labeled neurons.

**Source data 2.** Pixel-wise line intensity profiles of *pHlava-9E* labeled neurons _2.

**Source data 3.** O/R values along line profile.

**Source data 4.** Distributions of O/R values for 3 representative regions.

**Source data 5.** pH calibration profile of *pHlava-9E*.

**Figure supplement 1.** Design and characterization of *pHlava-9E*.

accurate to measure neutral or near neutral pH and well positioned to measure pH between pH 6 and 8 (*Figure 5f*). It is also designed such that any compartments with acidities ranging from pH 4 to 6 will all show the maximum O/R ratio. This is so that only two states are available for O/R values in order to assess how much of *pHlava-9E* is present on the surface versus in a compartment. Thus *pHlava-9E* gives an unambiguous readout of surface-immobilized to endocytosed fractions.

To estimate the amount of membrane bound versus vesicular labeling of DNA devices in p*snb-1::snb-1::9E* worms, we applied *pHlava-9E* to three nematode systems. p*snb-1::snb-1::9E* and p*snb-1::odr-2::9E* worms were injected with 1 µM of *pHlava-9E* (*Figure 5*). p*snb-1::odr-2::9E* worm neurons are expected to show negligible internalization of *pHlava-9E* and maximal plasma membrane labeling of ODR-2, a well-known neuronal cell-surface marker (*Chou et al., 2001*; *Gottschalk and Schafer, 2006*). In line with our expectations, the O/R ratio obtained, as revealed from the fold change in signal with respect to that in lysosomes, was 0.65 ± 0.03, corresponding to a pH of 7.2 (*Figure 5b,d*). The

O/R value of the punctate structures in *pHlava-9E* labeled neurons of p*snb-1::snb-1::9E* worms was 0.99 ± 0.03 (**Figure 5c,d**).

As a positive control for internalization, we imaged endosomes of the coelomocytes in p*snb-1::snb-1::9E* worms injected with *pHlava-9E* (**Figure 5a,d**). In this system, at the time of measurement, *pHlava-9E* is internalized and trafficked to lysosomes (**Chakraborty et al., 2017**). As expected, the O/R ratios obtained corresponded to the maximum possible value of 0.99 ± 0.03 reported by this device (**Figure 5d**). Based on the comparison of values obtained from the neurons of p*snb-1::snb-1::9E* worms, maximally 7 % of nanodevice could still be on the plasma membrane without nudging the O/R value beyond the experimental error (0.03) (**Figure 5f**). This indicates that minimally 93 % of *pHlava-9E* is present in compartments in the neurons of p*snb-1::snb-1::9E* worms.

## Conclusion

Endocytosis and endosome trafficking are complex processes, regulating many critical cellular functions and therefore play pivotal roles in tissue-specific physiology and pathology. Our knowledge of subcellular dynamics and organization of endocytic and recycling pathways stems largely from investigations in cultured cellular systems. However, in vivo, endocytosis in single cells and their functional outcomes are sculpted by molecular interactions with their neighboring cells, occurring in the presence of multiple biochemical cues originating from other tissues. Thus, the ability to study endocytic pathways in vivo, at the single-cell level and in their native context, would illuminate our understanding of this core cellular process at the whole organism level.

There is now a rich repertoire of investigative tools based on DNA nanodevices that either probe or manipulate endosomal trafficking pathways (**Krishnan et al., 2020**). However, in order to tap the full potential of DNA-based nanodevices in vivo, it is essential to develop strategies for precise targeting to specific tissues. Here, we leverage both endogenous as well as synthetic nucleic acid-binding receptors to specifically target simple DNA nanodevices to two kinds of tissues: IECs and neurons in *C. elegans*.

Our results show that, unlike what has been observed for coelomocytes (**Chakraborty et al., 2017**; **Dan et al., 2019**; **Narayanaswamy et al., 2019**; **Surana et al., 2011**; **Veetil et al., 2017**), simply introducing DNA nanodevices into the worm intestine and circumventing the relevant biological barrier does not lead to nanodevice targeting to IECs. This is due to the lack of an appropriate IEC surface receptor for DNA. However, by exploiting the endogenous dsRNA receptor SID-2 in IECs, we could target DNA nanodevices to these cells. We achieved this by conjugating a dsRNA domain to DNA nanodevices, where the dsRNA domain engaged the SID-2 receptor such that the DNA nanodevice underwent receptor-mediated endocytosis and was subsequently trafficked to LROs. Furthermore, since mutations in the *sid-2* gene, but not *sid-1*, abrogated DNA nanodevice uptake, we could pinpoint that nanodevice targeting occurs via the endogenous SID-2 receptor. Our results show that DNA nanodevices can be targeted tissue-specifically by exploiting relevant endogenously expressed cell-surface receptors and displaying the cognate ligand on the DNA nanodevice. Should new knowledge of cell type-specific nucleic acid receptors emerge, this strategy could be used to expand the targeting of DNA nanodevices to other classes of cells.

In the absence of such knowledge, we have developed a synthetic receptor-ligand strategy to target DNA nanodevices in a generalizable way so that it can be adapted to a variety of cellular and molecular contexts in vivo. We therefore identified a sequence-specific, DNA-binding recombinant humanized $V_H H$ antibody, denoted 9E, by phage display. 9E binds a specific 4-nt sequence, namely d(ATAA), when it is present at the 3' end of the DNA nanodevice with high affinity (~200 nM), irrespective of the environmental pH.

By fusing 9E to proteins that are expressed on the target cell under appropriate tissue-specific promoters, we can target DNA nanodevices with cellular- and subcellular-level precision. For example, since SNB-1 is ubiquitously expressed in all neurons, we expressed a SNB-1::9E fusion protein under the pan-neuronal promoter p*snb-1* and targeted a DNA nanodevice displaying the 3' terminal d(ATAA) sequence, specifically to neurons. Interestingly, nanodevices labeled endosomes not only at the synapses but all along the axon within the relevant neurons. Neuronal targeting required both SNB-1 fused to the 9E antibody and the DNA nanodevice to have the 3' d(ATAA) sequence. This established that cell-specific targeting was contingent upon the interaction between the synthetic receptor and its cognate DNA epitope. DNA nanodevices could be targeted to specific subsets of neurons by

expressing the SNB-1::9E chimera under promoters of genes expressed in specific neuronal classes. Thus, when SNB-1::9E was expressed under *pacr-2* or *pglr-4* promoters (*Feng et al., 2006*; *Kratsios et al., 2015*; *Kratsios et al., 2011*), DNA nanodevices were internalized by cholinergic ventral cord neurons or SAB head motor neurons, respectively.

We showed that targeting of DNA nanodevices could be achieved with subcellular-level precision using this strategy. Fusing 9E to the transmembrane odorant receptor, ODR-2, which does not undergo significant endocytosis (*Chou et al., 2001*), enabled the immobilization of a DNA nanodevice at the neuronal surface, rather than being endocytosed into intracellular vesicles. This suggests that DNA nanodevices may be localized with subcellular precision using a synthetic receptor and cognate ligand strategy in vivo. By using a functional DNA nanodevice, *pHlava-9E,* we showed that the punctate neuronal labeling observed in p*snb-1::snb-1::9E* worms is predominantly in internal compartments.

To interrogate organelle dynamics and function in vivo, the development of robust technologies with molecular specificity that enables access to discrete cell types is critical. The modular system described here has the capability to interface with and harness the power of nucleic acid nanotechnology to probe and modulate specific organelles in vivo. The methodology described here is confined to addressing those organelles that exchange membrane content with the plasma membrane, namely endocytic organelles. To target DNA nanodevices to organelles on the secretory pathway, such as the endoplasmic reticulum, new strategies that co-opt retrograde trafficking pathways are required. Further, this method currently cannot localize DNA nanodevices to the cytoplasm, the nucleoplasm, or within the mitochondria. Nevertheless, these targeting strategies open up an array of possible applications where DNA nanodevices can be readily applied in the multicellular context, positioning them to deliver key insights into complex biological phenomena.

# Materials and methods
## Oligonucleotides
All fluorescently labeled DNA oligonucleotides were HPLC-purified and obtained from IBA-GmBh (Germany) and IDT (Coralville, IA). Unlabeled DNA oligonucleotides were purchased from IDT. RNA was transcribed in vitro using a MEGAscript T7 Kit (Invitrogen, USA). geneBlock fragments were obtained from IDT.

Sequences of the various oligonucleotides used are listed in table in *Supplementary file 1*.

## Preparation of oligonucleotide samples

All oligonucleotides were ethanol precipitated, dissolved in Milli-Q water, aliquoted as a 100 μM stock, and stored at –20 °C. Concentration of each oligonucleotide was measured using UV absorbance at 260 nm.

An IEC targeting probe consists of an azido labeled RNA duplex conjugated to a dibenzocyclooctyl (DBCO) containing fluorophore or DNA duplex via copper-free click chemistry (*Baskin et al., 2007*; *Jewett et al., 2010*; *Paredes and Das, 2011*). A PCR fragment of 100 -bp or 50 -bp template with a T7 promoter site obtained from a plasmid DNA was used as a template for in vitro transcription (IVT) using protocols suggested by the manufacturer (MEGAscript T7 Kit, Invitrogen). The ability of T7 RP to incorporate different 5'-modified guanosine analogues was utilized in transcriptional priming to label and specifically conjugate the 5'-terminus of transcripts (*Huang et al., 2008*; *Milligan et al., 1987*) for incorporating a 5'-azide reactive group. In the IVT reaction, 5'-azido-5'-deoxy guanosine (5 '-$N_3$G) is added in fourfold excess over GTP to prime transcriptions (*Paredes and Das, 2011*). RNA formation was confirmed by gel electrophoresis (PAGE) (*Figure 2—figure supplement 1*). The transcript with the 5'-terminal azide (R-$N_3$) is subsequently used in click reactions with a fluorophore or DNA strand containing a DBCO group. The click conjugation is performed in 10 mM sodium acetate buffer pH 5.5. This conjugated strand was then annealed to the corresponding complementary strands heating at 90 °C for 5 min, and then slowly cooling to room temperature (RT) at 5 °C per 15 min.

A sample of duplex DNA for screening of DNA-binding $V_H$H was made by mixing the relevant DNA oligonucleotides in equimolar ratios, heating at 90 °C for 5 min, and then slowly cooling to RT at 5 °C per 15 min. This sample preparation was carried out with oligonucleotide concentrations of 5 μM, in

phosphate buffered saline (PBS) of pH 7.3, in the presence of 100 mM KCl. Samples were then equilibrated at 4 °C overnight. Samples were used within 7 days of annealing.

Selective oligonucleotides were phosphorylated at their 5' end by incubating them with T4 polynucleotide kinase (PNK; New England Biolabs, USA). 2 nmoles of the oligonucleotide were mixed with 2 µL of 10× PNK buffer, 2 µL of PNK (10 U/µL), 4 µL of 1 mM ATP, and the volume was made up to 20 µL. This reaction mix was incubated at 37 °C for 1 hr. Post-incubation, the enzyme was inactivated by incubating the mixture at 75 °C for 15 min. The DNA was subsequently ethanol precipitated, resuspended in Milli-Q water, and quantified using UV absorbance. These were then used to prepare duplex DNA using the protocol mentioned above.

A sample of the neuronal targeting probe (nD$^{A647N}$) was made by mixing the DNA oligonucleotides nD and nD' in equimolar ratios, heating at 90 °C for 5 min, and then slowly cooling to RT at 5 °C per 15 min. This sample preparation was carried out with oligonucleotide concentrations of 5 µM, in 10 mM phosphate buffer of pH 7.4, in the presence of 100 mM KCl. Samples were then equilibrated at 4 °C overnight.

## *pHlava-9E*: design and characterization

*pHrodoRed* (Thermo Fischer) was first conjugated to the ssDNA strand using a standard NHS coupling reaction. Briefly, 200 µM *pHrodoRed* succinimidyl ester (1 µL from a 10 mM stock in dry DMSO) was added to 20 µM NH$_2$-labeled ssDNA (D$_P$) in 100 µL of sodium phosphate buffer (10 mM) containing KCl (100 mM) at pH 7.0 and stirred overnight at RT. The reaction mixture was diluted to final volume of 1 mL using Milli-Q water. This solution was then centrifuged using a 3 kDa MWCO ultracentrifugation filter to remove unreacted *pHrodred*. The presence of *pHrodored* in the filtrate was negated by using UV-Vis spectroscopy ($l_{ab}$ = 560 nm, = 65,000 cm$^{-1}$ M$^{-1}$). The ratio of ssDNA to the conjugated *pHrodoRed* was confirmed to be 1:1 using UV-Vis spectroscopy. Conjugation of *pHrodoRed* was also confirmed using 15 % polyacrylamide gel electrophoresis (Supplementary information *Figure 5— figure supplement 1a* ). To prepare *pHlava-9E*, we mixed 10 µM D$_P$-*pHrodoRed* and ssDNA-ATTO 647N (D$_A$) strands in equimolar ratios in 10 mM sodium phosphate buffer (pH 7.2) containing 100 mM KCl. The resultant solution was heated to 90 °C for 15 min, cooled to RT at 5 °C per 15 min, and incubated at 4 °C for 8 hr. The *pHlava-9E* formation was verified using 15 % polyacrylamide gel electrophoresis (Supplementary information *Figure 5—figure supplement 1b*). The yield of *pHlava-9E* assembly was near quantitative.

## Preparation of helper phages

*Escherichia coli* strain TG1 grown in minimal media (M9 media) was inoculated 1:100 in 100 mL 2xTY media and grown till OD$_{600}$ reached 0.2. 200 µL of this culture was then transferred to tubes and infected with serial dilutions of 4 × 10$^{11}$ M13KO7 helper phage (with a minimum ratio of 1 bacterium per 20 helper phages, GE Healthcare, USA) (10$^{-10}$, 10$^{-11}$, 10$^{-12}$, 10$^{-13}$, 10$^{-14}$). The tube was quickly transferred to a 37 °C water bath and incubated for 30 min without shaking. 3 mL of H-Top agar was heated to 42 °C and quickly poured to each tube. This was then poured on 2xTY plates and incubated overnight at 37 °C. The plaques obtained from each plate were inoculated in 3 mL of 2xTY and grown till optical density at 600 nm (OD$_{600}$) reaches 0.5. At this stage, the cultures were again infected with helper phage and incubated for 2 hr at 37 °C with shaking. This was diluted in 500 mL of 2xTY media and incubated at 37 °C for 1 hr, after which kanamycin was added to a final concentration of 50 µg/mL and the culture was incubated overnight at 37 °C. The overnight culture was centrifuged at 10,800× g for 15 min, and the supernatant was collected into a glass conical flask. This flask was transferred to ice, and 100 mL ice-cold polyethylene glycol/NaCl solution (30 % PEG 8000 with 2.5 M NaCl) was added with constant shaking. The resulting mixture was kept on ice for 45 min. The solution became turbid and was centrifuged at 10,800× g for 30 min, and the supernatant was discarded. The phage precipitate was re-suspended in 6 mL PBS containing 30 % glycerol. This was filtered using a 0.45 µm membrane filter (Merck Millipore, USA), aliquoted, and stored at –80 °C.

## Preparation of library

Screening for DNA-binding sdAb was performed with biotinylated DNA oligonucleotides in native condition as described using Nali-H1 library composed of 3 × 10$^9$ synthetic humanized VHH (**Moutel et al., 2016**; **Nizak et al., 2005**). A 1 mL aliquot of the glycerol stock library (3 × 10$^{10}$ clones/mL)

was inoculated in 250 mL of 2xTY containing 100 µg/mL of ampicillin (Sigma-Aldrich, USA) and 1 % glucose. This was incubated at 37 °C till the $OD_{600}$ reached 0.5. 75 mL (containing 3.1010 clones) of this culture was infected with an excess of helper phage M13KO7, with a minimum ratio of 1 bacterium per 20 helper phages. Care was taken that no pipetting or shaking was done. This was incubated for 30 min at 37 °C, without agitation, in a water bath. The infected bacteria were centrifuged for 20 min at 4200 rpm, the pellet was re-suspended in 500 mL of 2xTY with 100 µg/mL ampicillin and 50 µg/mL kanamycin without glucose, and incubated overnight at 30 °C with shaking. The overnight culture was centrifuged at 10,800× g at 4 °C for 10 min. 100 mL of ice-cold PEG-NaCl solution was added to the supernatant of the centrifuged culture, such that the supernatant becomes cloudy. This was incubated for 1 hr on ice at 4 °C. The mixture was centrifuged for 30 min at 10,800× g at 4 °C to pellet the phages. The PEG/NaCl solution was aspirated off carefully without disturbing the pellet. The pellet was re-suspended in 40 mL Milli-Q water, after which 8 mL of PEG/NaCl was added. The solution was swirled for efficient but gentle mixing and then allowed to stand on ice at 4 °C for 20 min. The phages were centrifuged again at 10,800× g for 30 min. The PEG/NaCl supernatant was carefully removed. The pellet containing a purified library of phages was re-suspended in 5 mL cold PBS and centrifuged again for 10 min at 13,000 rpm at 4 °C to pellet cell debris and bacteria. This is used as input for the phage display screen. The rest is stored as aliquots at –80 °C.

### Preparation of DNA-conjugated magnetic beads

150 µL streptavidin-coupled magnetic Dynabeads (Invitrogen) were washed three times in PBS supplemented with 0.1 % Tween-20 (Sigma-Aldrich) (PBST). After each wash, beads were collected using a magnet. 15 µL of the B-RO3-CELL duplex (oligonucleotides Biotin-R-CELL + O3-CELL) was added to the washed magnetic beads and the volume of the reaction made up to 500 µL with PBST such that the final DNA concentration used for the screening is 50 nM. This was incubated on a rotator for 1 hr. After 1 hr, the DNA-coated beads are washed in PBST three times and then suspended in 150 µL of PBS. For each round of selection, 50 µL aliquots of this DNA-conjugated magnetic bead mix were used. The rest was stored at –20 °C for future use.

### Selection of dsDNA binders using phage display method

The phage display screen was carried out using a previously described protocol 10, with some variations. Non-specific binders (in particular, anti-streptavidin $V_H$Hs) were removed from the library by incubating $V_H$Hs displayed on the surface of phages with streptavidin-coated magnetic beads alone. 50 µL of streptavidin-coated magnetic beads were mixed with 100 µL of the library (this input contains $3 \times 10^{12}$ phages) and incubated for 30 min on a roller and another 30 min standing. Volume of this mix was made up to 1.5 mL with PBST supplemented with 0.6 % non-fat milk (PBSTM). After 1 hr, the beads containing streptavidin binders were pulled down using a magnet, while the supernatant containing the remainder of the library was added to 50 µL of the DNA-coated magnetic beads. This mix was incubated on a roller for 30 min and then left standing for 1.5 hr. After 2 hr, the beads were pulled down using a magnet, the PBSTM was removed, 1 mL of fresh PBSTM was added, and the beads were fully re-suspended. This mix was added to a 15 mL polypropylene tube, which was pre-blocked with PBSTM (2 % non-fat milk), and the volume made up to 10 mL using PBST. The beads were collected using a magnet for 5 min, the PBST was removed, and fresh PBST was added. This was repeated 20 times, with a change of polypropylene tube every five washes. After the last wash, 1 mL of 100 mM triethylamine (TEA) was added to re-suspend and dissociate the phages from the beads; this was transferred to a tube and rocked for 7 min, after which the beads were collected using a magnet. 500 µL of the supernatant was carefully added to 500 µL of 1 M Tris-Cl (pH 7.4) to neutralize it; the remaining 500 µL was put back on the roller again for 7 min and the above steps were repeated. Finally, 200 µL of 1 M Tris-Cl was added to the polypropylene tube and kept separately.

### Rescue and enrichment of possible DNA binders

The rescue and amplification of selected phage was done by infecting *E. coli* TG-one bacteria with eluted phage. 750 µL of the recovered phages were added to 9.25 mL of a TG1 culture (grown to an $OD_{600}$ of 0.5 in 2xTY media with 1 % glucose) and incubated at 37 °C for 30 min in a water bath without agitation. Simultaneously, 4 mL of the same TG1 culture was added to the last polypropylene tube

and the infection protocol was repeated. Both the cultures were pooled to get 14 mL of infected TG1 culture. Of this, three dilutions of 1 mL each were made, $10^{-1}$, $10^{-2}$, and $10^{-3}$, and two volumes of each, 10 µL and 100 µL, were spread on 2xTY plates containing 1 % glucose and 100 µg/mL ampicillin. These plates were used for calculation of phage output after the first round of selection. The remaining culture was spun down at 3300 rpm for 10 min at RT. The pellet was re- suspended in 1.5 mL 2xTY, which was equally divided and plated on three large 2xTY plates containing 1 % glucose and 100 µg/ mL ampicillin. All plates were grown overnight at 37 °C.

Colonies obtained on the large Petri dishes were scraped off using 6 mL of 2xTY supplemented with 30 % glycerol. This scraped culture constitutes the input for Round 2 of screening. An aliquot of this culture was added to 100 mL of 2xTY with 1 % glucose and 100 µg/mL ampicillin, such that $OD_{600}$ of the inoculated culture was 0.05. This was incubated at 37° C till $OD_{600}$ reaches 0.5. 10 mL was aliquoted into a fresh tube, helper phages were added (bacteria:helper phage = 1:20; at $OD_{600}$ = 0.5, a 10 mL culture of *E. coli* contains 4 × 109 bacteria) and the culture was incubated at 37° C in a water bath for 30 min without agitation. The culture was centrifuged at 3300 rpm for 10 min at RT, the pellet was re-suspended in 50 mL 2xTY with 100 µg/mL ampicillin and 50 µg/mL kanamycin (without glucose) and incubated overnight at 30 °C.

40 mL of the overnight culture was centrifuged at 10,800× g at 4 °C for 10 min. 8 mL of ice-cold PEG-NaCl solution was added to the supernatant of the centrifuged culture, such that the supernatant becomes cloudy. This was incubated for 1 hr on ice at 4 °C. The mixture was centrifuged for 10 min at 10,800× g at 4 °C to pellet the phages. The PEG solution was aspirated off carefully without disturbing the pellet. The pellet was re-suspended in 2 mL cold PBS without introducing air bubbles. The phages were centrifuged again at 10,800× g for 10 min. The supernatant was carefully removed. The pellet containing cell debris and bacteria was discarded, while the supernatant was used as input for Round 2 of selection. 500 µL dilutions of $10^{-9}$, $10^{-10}$, and $10^{-11}$ were also made with the recovered phages, introduced into TG1 bacteria (as described above) and spread on 2xTY plates containing 1 % glucose and 100 µg/mL ampicillin to calculate input. This was done for three rounds of selection. Negative selection against streptavidin was done at each stage of selection to completely eliminate anti-streptavidin $V_H$Hs.

After three rounds of screening, 80 clones each from Rounds 2 and 3 were selected for further characterization. Individual colonies were transferred to one well of a 2 -mL-deep 96-well plate (Greiner Bio-one, Germany) with 600 µL 2xTY containing 100 µg/mL ampicillin and 1 % glucose, and grown overnight with shaking at 37 °C. Glycerol was then added to a final concentration of 30 % to make a master plate that was stored at –80 °C.

## ELISA using phages

ELISA was performed using phages secreted in the media, each of which carries the $V_H$H fused to a coat protein. 600 µL of 2xTY with 100 µg/mL ampicillin and 1 % glucose were added to each well of 2 -mL-deep 96-well plates, each of which was inoculated with 6 µL of the master stock. This was incubated for 2.5 hr at 37 °C with shaking, such that $OD_{600}$ reaches 0.5. Helper phages were added to each well, and the plates were incubated at 37 °C without agitation. The plates were then centrifuged at 2500 rpm for 5 min. The supernatant was carefully aspirated off and the pellet re-suspended in 600 µL 2xTY containing 100 µg/mL ampicillin and 50 µg/mL kanamycin. The plates were incubated overnight at 30 °C with agitation. The phages were recovered by centrifuging the cultures at 2800 rpm for 10 min at RT and pipetting the supernatant in fresh 96-well plates.

In order to perform the ELISA, 96-well ELISA plates (Nunc Maxisorp, Thermo Fisher Scientific, USA) were coated with 50 µL of 20 µg/mL avidin and incubated at RT for 2 hr. The plates were flicked and washed once with PBS to remove excess avidin. 50 µL of biotinylated DNA at 50 nM concentration was immobilized by adding it to the wells and incubating at 4 °C overnight. Excess DNA was removed by flicking the plates. The plates were blocked using 200 µL of and kept at RT for 1 hr, after which it was removed by flicking. In a separate plate, 90 µL of phages were mixed with 20 µL of PBSTM, incubated at RT for 20 min, and then 100 µL of this mix was added to the ELISA plate. The phages and DNA were allowed to bind for 2 hr, after which the plates were flicked and washed three times each with PBST and PBS. 50 µL of anti-M13 antibody conjugated to horseradish peroxidase (HRP; GE Healthcare Life Sciences, USA) was added at a 1:5000 dilution and incubated for 40 min. The plates were flicked and washed three times with PBST and PBS. 100 µL of tetramethyl benzidine (TMB)/$H_2O_2$

(BD Biosciences, USA) was added to each well, and the reaction was stopped by addition of 100 μL of 2 N H$_2$SO$_4$. Binding was quantified by measuring absorbance at 450 nm using a Spectramax multi-mode plate reader (Molecular Devices, USA).

## ELISA using secreted V$_H$Hs

ELISA was performed using V$_H$Hs secreted in the media. 600 μL of 2xTY with 100 μg/mL ampicillin and 1 % glucose was added to each well of 2 -mL-deep 96-well plates, each of which was inoculated with 6 μL of the master stock. This was incubated for 2 hr at 37 °C with shaking, after which 1 mM isopropyl thiogalactoside (IPTG) was added. The plates were then shifted to 30 °C and incubated overnight. The cultures were centrifuged at 2800 rpm for 10 min at RT, and the supernatant, which contains secreted V$_H$Hs, was carefully transferred to a fresh 96-well plate. After this, standard ELISA, as outlined in the above protocol (ELISA using phages) was carried out, using mouse anti-Myc antibody (Sigma-Aldrich) at a dilution of 1:1500 and goat anti-mouse antibody conjugated to HRP (Life Technologies, USA) at a dilution of 1:1000.

## Characterization of sequence specificity

Approximately 80 colonies each from rounds 2 and 3 of selection were screened by ELISA against the dsDNA of choice (Biotin-R-CELL+ O3-CELL). This screening was done using both phage and protein ELISA, as described above. In order to check the sequence specificity of the positive clones, V$_H$Hs were subjected to another round of ELISA assay against various DNA epitopes. The epitopes used were ssDNA (Biotin-R-CELL, Biotin-O3-CELL), dsDNA (Biotin-R-CELL+ O3-CELL), divergent dsDNA (Biotin-DS1- CELL+ DS2-CELL), and various parts of the dsDNA (Biotin-R-CELL+ R1-CELL, Biotin-R- CELL+ RM-CELL, Biotin-R-CELL+ R2-CELL, Biotin-R-CELL+ R3-CELL, Biotin-R-CELL+ R4-CELL, Biotin-R-CELL+ R5-CELL). After immobilization of these targets, ELISA assay was carried out as mentioned above.

### Sequencing of specific clones
About 42 clones showing binding to dsDNA were chosen for sequencing. Individual clones were grown overnight and subjected to a plasmid DNA isolation using Nucleospin Plasmid Miniprep Kit (Macherey-Nagel GmbH, Germany). Sequencing was performed using a standard dideoxy sequencing method.

### Expression and purification of V$_H$Hs
V$_H$H expression was performed in M9 minimal media (1× M9 salts, 2 mM MgSO$_4$, 1 % glycerol, 0.1 % casamino acids, and 0.000005 % thiamine). Selected V$_H$H was inoculated in 100 mL 2xTY supplemented with 100 μg/mL ampicillin and 1 % glucose and grown overnight at 37 °C. 5 ml of this culture was inoculated in 500 mL of M9 minimal media with 100 μg/mL ampicillin and grown at 37 °C for 2 hr. IPTG was added to a final concentration of 1 mM, after which the culture was shaken at 30 °C for 16 hr. Bacteria were centrifuged at 10,000 rpm for 10 min. The supernatant containing secreted V$_H$Hs was filtered using a 0.22 μm membrane filter to remove the remaining debris. The filtered media was then incubated with 2 mL of Talon Cobalt affinity resin (Clontech Laboratories Inc, USA) for 1 hr at 4 °C. Post-binding, the media with the resin was put into a funnel adapted onto a column, which was pre-washed with 20 mL of PBS, at 4 °C. The flow through was collected and again passed through the column in order to collect all the metal beads with bound protein. This was done three times. Once all the beads were collected, they were washed again by passing 100 mL of PBS through the column. Non-specific binding to beads was abrogated using 1 mL 5 mM imidazole. The bound V$_H$H was then eluted using a gradient of imidazole concentrations, ranging from 50 mM to 250 mM. The first nine fractions were run on a 12 % SDS-PAGE to check the expression and elution of the protein.

Each eluted V$_H$H was further purified by removing imidazole using a 10 -kDa Amicon ultra centrifugal filter (Merck Millipore). Each fraction was added to the centrifugal filter and centrifuged for 10 min at 4 °C for 14,000 rpm. Volumes were then made up to 400 μL using PBS after each spin. This was repeated 10 times. Concentration of each fraction was measured using Bradford assay. Purified V$_H$H were stored at 4 °C for short term and at –20 °C for long term.

## pH-dependent ELISA

pH-dependent ELISA to check pH sensitivity of the selected $V_H$H was performed using purified $V_H$H. The $V_H$H solution was divided into two pools, and each pool was incubated with PBSTM of the desired pH for 20 min. This was then added to ELISA plates containing the immobilized DNA of choice. Standard ELISA, as described above, was then employed to assess binding.

## Determination of $V_H$H-binding affinities

Affinity of selected $V_H$Hs for their dsDNA epitopes was assessed using three formats, as described below. All data was analyzed using OriginPro 8.5 (OriginLab, USA).

i.   Using serial dilutions of immobilized DNA: Avidin-coated 96-well plates were incubated with serial dilutions of the dsDNA antigen (10 nM - 5 µM), . 250 nM of the protein was added and allowed to bind for 2 hr. ELISA was then carried out as described above.

ii.  Using serial dilutions of competitive DNA: 500 nM of the biotinylated dsDNA antigen was mixed with serial dilutions of competitor non-biotinylated dsDNA (1 nM to 25 µM), to which 100 nM of protein was added. This mixture was allowed to incubate for 2 hr at RT, after which it was added to avidin-coated 96-well plates and further incubated for 2 hr. ELISA was then performed.

iii. Using serial dilutions of purified $V_H$H: Avidin-coated 96-well plates were incubated and bound with 500 nM of the dsDNA antigen as described under ELISA using phages section. Serial dilutions of the protein (1 nM to 5 µM) were added and allowed to bind for 2 hr. ELISA was then performed.

## Electrophoretic mobility shift assay

The required dsDNA constructs were annealed, as described above, at concentrations of 5 µM. A binding reaction was setup consisting of 1 µM DNA and 1 µM protein in PBS. Glycerol was added to a final concentration of 10 % in order to minimize DNA-protein complex dissociation. DNA-protein complexes were allowed to form at RT for 2 hr, after which 50 pmoles of DNA were loaded on an 8 % native PAGE. The gel was run at 4 °C using Tris-acetate-EDTA (TAE) buffer at 100 V. The DNA-protein complexes were visualized using ethidium bromide staining.

## Plasmid vectors and construction of 9E fusions

PHA-1 plasmid (gift from Krastosis lab, University of Chicago) and pBluescript SK(-) (Agilent Technologies, USA) were used during construction of transgenic strains. PCR fragments containing promoters of choice, snb-1 or odr-2 and 9E, were generated using standard cloning protocols. psnb-1::odr-2 and snb-1::9E geneBlock fragments were obtained from IDT. pglr-4, psnb-1, 9E was cloned out of plasmids generated in house (pglr-4- Krastosis lab; psnb-1 and 9E- Krishnan lab). pacr-2 was cloned out of genomic DNA isolated from wild-type worms. All the PCR fragments had a unc-54–3'UTR sequence on their 3' end for better expression in worms (*Merritt et al., 2008*).

## *C. elegans* methods and strains

Wild-type strain used was the *C. elegans* isolate from Bristol (strain N2; *Brenner, 1974*). Mutant strains used are RRID:WBStrain00036339 VC1119 (*dyf-2&ZK520.2(gk505) III*), RRID:WBStrain00036661 VC1521 (*dyf-2(gk678) III*), RRID:WBStrain00035219 HC196 (*sid-1(qt9) V*). Other transgenics used for colocalization studies are as follows:

1. RRID:WBStrain00029076 *jsIs682* [p*rab-3::gfp::rab-3*] (*Bounoutas et al., 2009*; *Mahoney et al., 2006*), which expresses GFP::RAB-3 in all neurons.
2. RRID:WBStrain00029619 *otIs355* [*rab3p(prom1)::2xNLS::TagRFP*], which expresses RFP in the nucleus of all neurons (*Nguyen et al., 2016*).
3. RRID:WBStrain00000189 *otIs45* [*unc-119::GFP*], which expresses GFP in the cytosol of all neurons (*Altun-Gultekin et al., 2001*).
4. RRID:WBStrain00033470 *pwIs50* [*lmp-1::GFP+ Cbr-unc-119(+)*], which expresses LMP-1::GFP, a lysosomal marker (*Treusch et al., 2004*).
5. RRID:WBStrain00040193 *hjIs9* [*ges-1p::glo-1::GFP + unc-119(+)*] in which GFP is targeted to LROs in intestinal cells (*Zhang et al., 2010*).

All strains were procured from the Caenorhabditis Genetics Centre (CGC; University of Minnesota, USA). Standard methods were followed for the maintenance of *C. elegans*.

Transgenic strains were created by co-injecting PCR fragment of the gene of interest and PHA-1 plasmid (20 ng/μL) as a selectable marker in *pha-1* mutants (*Mello et al., 1991*). The PCR fragments used in this study were p*snb-1::snb-1::9E* p*snb-1::odr-2::9E*; p*glr-4::snb-1::9E* and p*acr-2::snb-1::9E*. Total concentration of injected DNA was 100 ng/μL, which was achieved using empty pBluescript SK(-) vector. Transgenic lines obtained were also maintained using standard protocols. *pha-1* mutants are temperature-sensitive embryonic lethal mutants that can grow at 15 °C but not 25 °C (*Haberland and Fogelman, 1985*). Each line was characterized based on the survival at 25 °C.

## Intestinal epithelia and neuron labeling

Intestinal labeling was done by soaking worms in a 10 μL, 1 μM solution of intestine targeting probe for 2 hr. Worms were then washed in PBS and placed on OP50 plates for another hour before imaging. Worms were mounted on 2.0 % agarose pads and anesthetized using 40 mM sodium azide in M9 buffer.

Neuronal labeling was done by injecting 500 nM of nD$^{A647N}$. Injections were performed, as previously described (*Stinchcomb et al., 1985*), in the dorsal side in the pseudocoelom, just opposite the vulva, of 1-day-old adult wild-type hermaphrodites. Injected worms were mounted on 2.0 % agarose pads and anesthetized either using 40 mM sodium azide or 1 mM levimasole in M9 buffer. Neuronal labeling was examined after 30–60 min of incubation at 22 °C. Imaging of labeled neurons was carried out in at least 15 worms.

## Colocalization experiments

For gut colocalization experiments, ~10 one-day-old adult transgenic worms were incubated in a solution of 1 μM R$^{50}$D$^{38}$ for 2 hr. Worms were then washed and incubated on OP50 plates for 1 hr for the clearing of the intestine lumen of excess sensor.

For neuronal colocalization experiments, p*snb-1::snb-1::9E* were crossed with various transgenic lines containing fluorescently labeled neuronal markers. Briefly, L4 for hermaphrodite worms of various fluorescently labeled transgenic lines were crossed with N2 males and the male progeny containing a fluorescent label were selected for the next step. Fluorescently labeled male worms were crossed with p*snb-1::snb-1::9E* containing L4 hermaphrodites. 10–12 hermaphrodite progeny from this step were singled out at the L4 developmental stage based on the presence of a fluorescent marker. These worms were allowed to self-fertilize and five of their progenies were used to perform single-worm PCRs to identify which plates contain worms with both 9E and the fluorescent neuronal markers. Single-worm PCRs were performed using standard methods (*Single worm PCR, 2019*). Briefly, five worms from each plate were placed into a tube containing 5 μL of worm lysis buffer (WLB: 50 mM KCl, 10 mM Tris pH 8.3, 2.5 mM MgCl$_2$, 0.8 % Tween-20, 0.01 % gelatin) and 100 μg/mL proteinase K. Worms are frozen at –80 °C for at least 1 hr. The tube is immediately treated at 60 °C for 60 min after which proteinase K is inactivated by heating to 95 °C for 15 min. The PCR mix is then added directly to the digested heat-inactivated sample. The PCR product was then confirmed by gel electrophoresis for 9E. Actin was used as a control. Plates that showed the presence of p*snb-1::snb-1::9E* were taken forward for injecting the DNA device similar to the protocols mentioned above.

Worms were mounted on 2.0 % agarose pads and anesthetized using either 40 mM sodium azide or 1 mM levimasole in M9 buffer and imaged on Leica TCS SP5 II STED laser scanning confocal microscope (Leica Microsystems, Inc, Buffalo Grove, IL) .

## Quantification of colocalization

For colocalization studies of R$^{50}$D$^{38}$ with various GFP-labeled LRO markers in the intestine, Pearson's correlation coefficient was calculated using *Coloc2* plugin on *ImageJ* (RRID:SCR_003070). In neurons, colocalization between nD$^{647}$ and *rab3::GFP* was obtained by plotting the z profile for lines drawn perpendicular to the neuron. Percentage colocalization in neurons was calculated as number of red puncta colocalized with green (obtained from the line profile) out of the total number of red puncta along the neuron.

## Quantification of neuronal pH using *pHlava-9E*

For the in vitro calibration curve, 200 nM of *pHlava-9E* dissolved in 20 mM pH clamping buffer of a given pH (120 mM KCl, 5 mM NaCl, 1 mM $CaCl_2$, 1 mM $MgCl_2$, 20 mM HEPES and 20 mM MES) was excited at 560 nm (for *pHrodoRed* emission) and 640 nm (for ATTO 647 emission). The emission spectra for *pHrodoRed* (O) and ATTO 647 (R) were collected from 570 to 620 nm and 650–750 nm, respectively. The ratio of fluorescence intensity at 580 nm and 660 nm were plotted as a function of pH to obtain a calibration curve. For the neuronal pH measurements, *pHlava-9E* (1 µM) was microinjected in the dorsal side in the pseudocoelom, just opposite the vulva, of 1-day-old hermaphrodites. Injected worms were then placed in a new nematode growth medium (NGM) agar-containing Petri plate at room temperature for 1 hr for maximum labeling of neurons and coelomocyte lysosomes. Worms were then mounted on an agar pad (2.0%) and anesthetized using 40 mM sodium azide in M9 buffer and imaged using wide-field microscopy.

## Wide-field microscopy and image analysis

Wide-field microscopy was performed on an IX83 inverted microscope (Olympus) using a ×60, 1.42 NA, phase-contrast oil immersion objective (PLAPON, Olympus) and Evolve Delta 512 electron-multiplying charge-coupled device (EMCCD) camera (Photometrics). The filter wheel, shutter, and CCD camera were controlled using Metamorph Premier Version 7.8.12.0 (Molecular Devices) (RRID:SCR_002368) as appropriate for the fluorophore used. Images were acquired on the same day under the same acquisition settings. Images were analyzed with ImageJ ver 1.49 (NIH, USA) (RRID:SCR_003070). All images were background subtracted using the mean intensity calculated from an adjacent cell-free area. The regions of cells containing neurons/lysosomes in each Alexa 647 (R) image were identified and marked in the ROI plugin in ImageJ (RRID:SCR_003070). The same regions were identified in the TMR image recalling the ROIs and appropriate correction factor for chromatic aberration if necessary. The ROI for neuron was selected by drawing a line with a length of 5 µm parallel to the maximum intensity region of the neurons. Mean fluorescence intensity was measured in TMR (O) and Alexa 647 channels. The ratio of the O/R intensities was calculated from these values. The mean O/R obtained from the coelomocytes (internal calibrator) was taken as the value corresponding to pH 5.5. The fold change in the O/R value relative to this value calculated from the calibration plot was used for the determination of pH of neurons.

## Confocal microscopy and image analysis

Confocal images were captured with a Leica TCS SP5 II STED laser scanning confocal microscope (Leica Microsystems, Inc) (RRID:SCR_018714) equipped with 63× , 1.4 NA, oil immersion objective. Alexa 488 was excited using an Argon ion laser for 488 nm excitation, Alexa 647 using He-Ne laser for 633 excitation. Images on the same day were acquired under the same acquisition settings. All the images were background subtracted prior to any image analysis, which was carried out using ImageJ ver.1.49d (NIH).

Image analysis for quantification of uptake during intestinal labeling was done using custom MATLAB code. For each worm, the most focused plane was manually selected in the Alexa 647 channel. To determine the location of the endosome, first a low threshold was used to select the entire field. Only the area within the cell was subsequently considered for vesicle selection. ROIs corresponding to individual vesicle were selected in the Alexa 647 channel by adaptive thresholding using Sauvola's method (*Sauvola and Pietikäinen, 2000*). The initial selection was further refined by watershed segmentation and size filtering. After segmentation, ROIs were inspected in each image and selection errors were corrected manually. Using this, we measured the mean fluorescence intensity for ~500 vesicles from ~10 animals and the background intensity corresponding to that field was subtracted.

## Acknowledgements

We thank Verenice S Noyola for assistance in drawing schemes for figures, and Anand Saminathan for his valuable comments related to functional nanodevice application and analysis. We thank the Integrated Light Microscopy facility at the University of Chicago and the Caenorhabditis Genetic Center (CGC) funded by NIH Office of Research Infrastructure Programs (P40 OD010440) for strains. This

work was supported by the University of Chicago Women's Board; FA9550-19-0003 from AFOSR (YK), NIH grants 1R01NS112139-01A1 (YK), the Ono Pharma Foundation Breakthrough Science Award (YK), and a Whitehall Foundation Grant 2017-12-50 (PK). The work in FP lab was supported by the Agence Nationale de la Recherche (ANR) and the Labex Cell(n)Scale (ANR-11-LABX-0038) part of ANR-10-IDEX-0001-02 PSL.

## Additional information

### Competing interests
Yamuna Krishnan: Reviewing editor, *eLife*. The other authors declare that no competing interests exist.

### Funding

| Funder | Grant reference number | Author |
|---|---|---|
| Air Force Office of Scientific Research | FA9550-19-0003 | Yamuna Krishnan |
| National Institute of Neurological Disorders and Stroke | 1R01NS112139-01A1 | Yamuna Krishnan |
| Ono Pharmaceutical | | Yamuna Krishnan |
| Whitehall Foundation | | Paschalis Kratsios |
| Agence Nationale de la Recherche | ANR-10-IDEX-0001-02 PSL | Franck Perez |
| Labex Cell(n)Scale | ANR-11-LABX-0038 | Franck Perez |

The funders had no role in study design, data collection and interpretation, or the decision to submit the work for publication.

### Author contributions
Kasturi Chakraborty, Data curation, Formal analysis, Investigation, Methodology, Project administration, Resources, Validation, Validation, Visualization, Writing – original draft, Writing – review and editing; Palapuravan Anees, Data curation, Formal analysis, Investigation, Methodology, Resources, Validation, Writing – original draft, Writing – review and editing; Sunaina Surana, Data curation, Formal analysis, Investigation, Methodology, Resources, Visualization, Writing – original draft, Writing – review and editing; Simona Martin, Resources; Jihad Aburas, Methodology, Resources; Sandrine Moutel, Investigation, Methodology, Resources; Franck Perez, Investigation, Methodology, Resources, Validation; Sandhya P Koushika, Funding acquisition, Writing – review and editing; Paschalis Kratsios, Funding acquisition, Project administration, Resources, Validation, Writing – review and editing; Yamuna Krishnan, Funding acquisition, Funding acquisition, Methodology, Project administration, Resources, Validation, Writing – original draft, Writing – review and editing

### Author ORCIDs
Kasturi Chakraborty (iD) http://orcid.org/0000-0002-0635-9028
Palapuravan Anees (iD) http://orcid.org/0000-0003-2118-8893
Sunaina Surana (iD) http://orcid.org/0000-0002-7017-3105
Sandrine Moutel (iD) http://orcid.org/0000-0002-9585-4855
Franck Perez (iD) http://orcid.org/0000-0002-9129-9401
Sandhya P Koushika (iD) http://orcid.org/0000-0002-1742-7356
Paschalis Kratsios (iD) http://orcid.org/0000-0002-1363-9271
Yamuna Krishnan (iD) http://orcid.org/0000-0001-5282-8852

### Decision letter and Author response
Decision letter https://doi.org/10.7554/eLife.67830.sa1
Author response https://doi.org/10.7554/eLife.67830.sa2

## Additional files

### Supplementary files

• Supplementary file 1. Table listing all oligonucleotides used in this study.

• Transparent reporting form

### Data availability

All data generated or analysed during this study are included in the manuscript and supporting files.

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
