## [Decision Letter]

**Acceptance summary:**

The editors and reviewers concur that targeting extraneously introduced DNA nanodevices to specific cell types with sub-cellular precision is exciting. The amenability of DNA nanostructures to tissue-specific targeting in vivo significantly expands their utility in biomedical applications and discovery biology.

**Decision letter after peer review:**

Thank you for submitting your article "Tissue specific targeting of DNA nanodevices in a multicellular living organism" for consideration by *eLife*. Your article has been reviewed by 2 peer reviewers, and the evaluation has been overseen by a Reviewing Editor and Piali Sengupta as the Senior Editor. The reviewers have opted to remain anonymous.

The reviewers have discussed their reviews with one another, and the Reviewing Editor has drafted this to help you prepare a revised submission. We all felt that this manuscript is interesting and is potentially suitable for publication in *eLife*. However both reviewers identified similar major revision request.

Essential revisions:

1) There needs to be a demonstration of the utility of this method. For example, linking the dsDNA to a pH sensitive dye to label synaptic vesicle, or any other "payload".

2) Quantitation of SNB-1 membrane vs internal staining will help to make the paper stronger.

*Reviewer #1 (Recommendations for the authors):*

The Krishnan group has developed many dsDNA devices with greater utility than D38, such as those that can report on enzymatic activity, pH, ion concentration, etc. A demonstration that the 9E targeting approach works with one of these biosensors would more directly show the general utility of the new approach. While fluorescence intensity data was well quantified, quantification of colocalization data by PCC or similar methods was lacking. Something appears wrong with the unc-119::gfp labeling in Supp Figure 7b. The GFP labeling looks like muscle not neurons.

*Reviewer #2 (Recommendations for the authors):*

1. I was anticipating some sort of functional demonstration in either the intestinal or neuronal delivery example perhaps based on other successful dsDNA probes utilized in the worm by the authors in previous studies (Devany 2018, Narayanaswamy 2019, Dan 2019, Chakraborty 2017 to name a few). Since this current manuscript only shows the fate of the Alexa dye, one wonders if the dsDNA is stable and functional under these conditions. The authors may be convinced that this is a non-issue, but they should either provide a bit of experimental evidence or a more convincing discussion of this point.

2. While the colocalization of the A647 dye and LMP-1::GFP appeared convincing from the example shown in Supplemental Figure 1, the GLO-1::GFP colocalization was much less convincing. The data is anecdotal without providing some sort of quantification such as the percentage of LMP-1 and/or GLO-1 positive LROs containing A647 dye signal (perhaps buffer control vs R100D38). Even something simplistic like Pearson correlation would help support the claim that the dsDNA was successfully delivered specifically to LROs. A similar criticism of the RAB-3 colocalization in Figure 4 could be made as well. Presumably this can be done with images the authors have already collected and would not require any new experiments.

3. I was impressed by the SNB-1::9E experiment and even without image quantification, it is clear that the authors have successfully recruited a dsDNA probe to neurites. Because there is a significant fraction of SNB-1 on the neuronal plasma membrane at steady state (both endogenous and over-expressed), and this surface SNB-1 is both synaptic and extrasynaptic, the dsDNA capture is likely to occur over the entire surface of neurons expressing the nanobody. Although the A647 dye signal looks somewhat punctate, we don't know whether this is surface or internal signal. Nor do we know with any quantitative precision where these enrichments are relative to synapses. Even plasma membrane proteins can appear to have a somewhat punctate distribution due to geometric effects (like the larger surface area of boutons) and surface lipid/protein heterogeneity. I think the claims made in this part of the manuscript are a bit over-stated since the degree to which the signal can be localized to subcellular compartments or even to somewhere besides the surface of the neurite is limited and unconvincing. All of this could be better handled with improved image analysis and some discussion of the these complicating factors. Perhaps FRAP experiments or pH-sensitive probes could clarify surface versus internal pools, but I do not think these experiments are necessary for publication. The authors also claim that some of the puncta are retrogradely transported vesicles. Is this anecdotal? Is there evidence provided for this claim in the manuscript? Given that SNB-1::9E decorates the entire neuronal surface, one could imagine internalized A647 would not necessarily be synaptic to begin with regardless of the transport fate of those compartments.

---

## [Author Response]

Essential revisions:1) There needs to be a demonstration of the utility of this method. For example, linking the dsDNA to a pH sensitive dye to label synaptic vesicle, or any other "payload".

We used a DNA nanodevice, denoted *pHlava*-*9E*, that uses pHrodo as a pH-sensitive dye. *pHlava*-*9E* is designed to provide a digital output of compartmentalization i.e., its pH profile is such that even if it is internalized into a mildly acidic vesicle, the pH readout is as high as one would observe with a lysosome. This gives an unambiguous readout of surface-immobilized probe to endocytosed probe.

2) Quantitation of SNB-1 membrane vs internal staining will help to make the paper stronger.

We applied *pHlava-9E* to three nematode systems. The in vitro pH performance curve of *pHlava*-*9E* shows a maximum fold change in O/R of 2.5 from pH 8 to pH 6.0, indicating that *pHlava*-*9E* is particularly accurate to measure neutral or near neutral pH and well positioned to measure pH between pH 6 to 8. It is also designed such that any compartments with acidities ranging from pH 4 to 6, will all show the maximum O/R ratio. This is so that only two states are available for O/R values in order to assess how much of *pHlava*-*9E* is present on the surface, versus in a compartment.

As a positive control for internalization, we imaged endosomes of the coelomocytes in p*snb-1:snb-1::9E* worms injected with *pHlava*. In this system, at the time of measurement, *pHlava*-*9E* is internalized and trafficked to lysosomes et al.(Chakraborty , 2017). As expected, the O/R ratios obtained corresponded to the maximum possible value of 0.99±0.028 reported by this device.

As a negative control, O/R ratios of *pHlava*-*9E* injected into p*snb-1:odr-2::9E* worms, is expected to label only the surface of the neuronal cell membrane. The O/R ratio obtained, as revealed from the fold change in signal with respect to that in lysosomes was 0.65±0.032, corresponding to a pH of 7.2.

When we compared the O/R value of the punctate structures in *pHlava*-*9E* labeled neurons of p*snb-1:snb-1::9E* worms, we observed a value of 0.99±0.03. This means that maximally 7% of nanodevice could still be on the plasma membrane without nudging the O/R value beyond the experimental error (0.03). This indicates that minimally, 93% of *pHlava*-*9E* is present in compartments.

Further, antibody-based trafficking studies by other groups, of retrogradely moving snb-1-containing compartments showed little or no surface labeling which could be reversed in mutants with reduced endocytosis et al.(Murthy , 2011).

Reviewer #1 (Recommendations for the authors):The Krishnan group has developed many dsDNA devices with greater utility than D38, such as those that can report on enzymatic activity, pH, ion concentration, etc. A demonstration that the 9E targeting approach works with one of these biosensors would more directly show the general utility of the new approach.

Please see point 1 of “Essential Revisions”.

While fluorescence intensity data was well quantified, quantification of colocalization data by PCC or similar methods was lacking.

We have now quantified colocalization in each cellular system. Please see Figure 2—figure supplement 1E and Figure 4—figure supplement 2E of the revised manuscript.

Something appears wrong with the unc-119::gfp labeling in Supp Figure 7b. The GFP labeling looks like muscle not neurons.

The reviewer is correct. Actually, it is known that *unc-119* is expressed in neurons and muscles and is also described in WormBase: https://wormbase.org/species/c_elegans/gene/WBGene00006843#0-9f1d4ihc-10

Earlier, we showed representative images of DNA nanodevices labeling the head due to the abundance of neurons there. We now also show other representative images of neurons along the worm body (Figure 4—figure supplement 2C).

Reviewer #2 (Recommendations for the authors):1. I was anticipating some sort of functional demonstration in either the intestinal or neuronal delivery example perhaps based on other successful dsDNA probes utilized in the worm by the authors in previous studies (Devany 2018, Narayanaswamy 2019, Dan 2019, Chakraborty 2017 to name a few). Since this current manuscript only shows the fate of the Alexa dye, one wonders if the dsDNA is stable and functional under these conditions. The authors may be convinced that this is a non-issue, but they should either provide a bit of experimental evidence or a more convincing discussion of this point.

We completely agree. Please see point 1 of “Essential Revisions”.

2. While the colocalization of the A647 dye and LMP-1::GFP appeared convincing from the example shown in Supplemental Figure 1, the GLO-1::GFP colocalization was much less convincing. The data is anecdotal without providing some sort of quantification such as the percentage of LMP-1 and/or GLO-1 positive LROs containing A647 dye signal (perhaps buffer control vs R100D38). Even something simplistic like Pearson correlation would help support the claim that the dsDNA was successfully delivered specifically to LROs.

Please see our last two responses to Reviewer 1.

In addition, we clarify the issue with GLO-1::GFP colocalization*.* We kept uniform washing times throughout for all the strains used in this study. At these washing times, GLO-1::GFP worms showed lower clearance of the intestinal lumen compared to all the other strains. Hence, most Glo-1::GFP images showed residual fluorescence in their lumens due to uncleared nanodevices. We now provide another representative image (Author response image 1) where the lumen is not visible due to the orientation of the worm, which shows nanodevice uptake more clearly.

**Author response image 1. sa2fig1:** a) Representative images showing colocalization between LRO marker, GLO-1::GFP and R^50^D^38^ nanodevices uptaken by intestinal epithelial cells b) Pearson’s correlation coefficient (PCC) calculated of the colocalization between R^50^D^38^ (red) and GLO-1 (green) in GLO-1::GFP worms.

A similar criticism of the RAB-3 colocalization in Figure 4 could be made as well. Presumably this can be done with images the authors have already collected and would not require any new experiments.

Please see our second response to Reviewer 1.

3. I was impressed by the SNB-1::9E experiment and even without image quantification, it is clear that the authors have successfully recruited a dsDNA probe to neurites. Because there is a significant fraction of SNB-1 on the neuronal plasma membrane at steady state (both endogenous and over-expressed), and this surface SNB-1 is both synaptic and extrasynaptic, the dsDNA capture is likely to occur over the entire surface of neurons expressing the nanobody. Although the A647 dye signal looks somewhat punctate, we don't know whether this is surface or internal signal. Nor do we know with any quantitative precision where these enrichments are relative to synapses. Even plasma membrane proteins can appear to have a somewhat punctate distribution due to geometric effects (like the larger surface area of boutons) and surface lipid/protein heterogeneity. I think the claims made in this part of the manuscript are a bit over-stated since the degree to which the signal can be localized to subcellular compartments or even to somewhere besides the surface of the neurite is limited and unconvincing. All of this could be better handled with improved image analysis and some discussion of the these complicating factors. Perhaps FRAP experiments or pH-sensitive probes could clarify surface versus internal pools, but I do not think these experiments are necessary for publication.

We are grateful to this reviewer for raising an excellent point, addressing which significantly improved our manuscript. We designed a nanodevice denoted *pHlava*, which reports on internalized nanodevices versus surface-displayed nanodevice.

We included a new Figure 5 in our manuscript describing a series of experiments with controls to show that surface labeling upon binding snb-1::9E in neurons is negligible.

The authors also claim that some of the puncta are retrogradely transported vesicles. Is this anecdotal? Is there evidence provided for this claim in the manuscript? Given that SNB-1::9E decorates the entire neuronal surface, one could imagine internalized A647 would not necessarily be synaptic to begin with regardless of the transport fate of those compartments.

See response to the point above. The puncta in which DNA devices are internalized are acidic. Further, fluorescently labeled anti-GFP has been shown to be enriched in compartments that are retrogradely transported where the Koushika group used labeled Anti-GFP which bound to SNB-1::GFP et al.(Murthy , 2011).

We now drop the claim that vesicles containing DNA nanodevices are retrogradely transported despite observing these vesicles moving retrogradely. This is because we feel that adding more data to demonstrate retrograde transport would distract from the main message of the paper – that DNA nanodevices can be targeted tissue-specifically and with sub-cellular precision.